**Public Perceptions of a Rip Current Hazard Education Program: 'Break the Grip of the**
**Rip!'**

Chris Houser[1], Sarah Trimble[2], Robert Brander[3], B. Chris Brewster[4], Greg Dusek[5] Deborah
Jones[6], John Kuhn[5]
[1]Department of Earth and Environmental Sciences
University of Windsor
401 Sunset Avenue
Windsor, Ontario, Canada
N9B 3P4
[2]Department of Geography
Texas A&M University
810 O&M Building
College Station, Texas, USA
18 77843-3148
[3]School of Biological, Earth and Environmental Sciences
UNSW Sydney
Sydney, NSW 2052 Australia
[4]United States Lifesaving Association
[5]NOAA/National Ocean Service
Center for Operational Oceanographic Products and Services
1305 East West Highway, SSMC4 #6636
Silver Spring, MD 20910
[6]National Weather Service
Marine, Tropical, and Tsunami Services Branch
**Abstract**
Rip currents pose a major global beach hazard; estimates of annual rip current related deaths in
the United States alone range from 35 to 100 per year. Despite increased social research into
beach-goer experience, little is known about levels of rip current knowledge within the general
population. This study describes results of an online survey to determine the extent of rip current
knowledge across the United States, with the aim of improving and enhancing existing beach
safety education material. Results suggest that the US-based "Break the Grip of the Rip"®
campaign has been successful in educating the public about rip current safety directly or
indirectly, with the majority of respondents able to provide an accurate description of how to
escape a rip current.  However, the success of the campaign is limited by discrepancies between
personal observations at the beach and rip forecasts that are broadcasted for a large area and
time.  It was the infrequent beach user that identified the largest discrepancies between the
forecast and their observations.  Since infrequent beach users also do not seek out lifeguards or
take the same precautions as frequent beach users, it is argued that they are also at greatest risk
of being caught in a dangerous situation. Results of this study suggest a need for the national
campaign to provide greater focus on locally specific and verified rip forecasts and signage in
coordination with lifeguards, but not at the expense of the successful national awareness
program.
**KEYWORDS**: Rip Current, Beach Safety, Survey, Perceived Risk

## 1 Introduction

Rip currents (often called "rips" or "rip tides") are strong, narrow seaward flows driven by alongshore variations in wave set-up landward of the breaker zone. Due to their dependence on wave breaking, rips can develop in any beach environment in oceanic, sea and lacustrine environments. Castelle et al. (2016) classify rips as: 1) boundary rips that develop along both natural and engineered structures including headlands, groins and piers, 2) bathymetric rips that develop in response to the variability of the nearshore morphology and 3) hydrodynamic rips that are spatial and temporally variable and develop in the absence of morphological variations or a lateral boundary. The type of rip that develops on a beach depends on the local wave climate and geology. For example, rips in the Great Lakes tend to be associated with natural headlands or the presence of large groins or harbor jetties, while rips in Florida and Texas tend to be bathymetrically controlled and associated with a transverse bar and rip nearshore morphology (Houser et al. 2013). Rips also vary regionally based on the driving forces, with rips on the Great Lakes typically associated with moderate to strong winds, while on the West Coast of the United States the rips are often associated with large swell events independent of the wind.

Rips are capable of carrying unsuspecting bathers significant distances away from the shoreline with speeds reaching over 2 m s$^{-1}$. As a consequence rips are considered a major public health problem in the USA, Australia, Costa Rica, and many other countries (Klein et al., 2003; Hartmann, 2006; Sabet and Barani, 2011; Woodward et al., 2013; Arun Kumar and Prasad, 2014). Rip currents in these countries are considered a major public health problem (Short and Hogan 1994; Sherker et al., 2008; Morgan et al. 2009; Arozarena et al., 2015). In Australia, rip currents are believed to be responsible for approximately 13,000 beach rescues per year (SLSA, 2016) and an average of 21 confirmed deaths per year (Brighton et al., 2013), which exceeds

fatalities caused by most other natural hazards (Brander et al., 2013). While it has been estimated
that 30–40 individuals drown each year in  rip current related incidents in the United States
(Gensini and Ashley 2010), Lushine (1991) suggested that rips may account for up to 150 fatal
drownings per year and the United States Lifesaving Association (USLA) estimate this number
to be over 100 per year. USLA's National Lifesaving Statistics Report (2012) indicates that over
82% of surf beach rescues in the US are rip current related and they therefore surmise that 82%
of all fatal drownings at beaches are associated with rip currents.
Beach users' vulnerability to drowning in a rip current depends on a combination of
nearshore hydrodynamic and bathymetric conditions, personal and group behaviors, and the
beach safety and rip current knowledge of the individual (e.g. Houser et al., 2011; Brander et al.,
2011; Caldwell et al., 2013; Houser et al., 2016). Morgan et al. (2009) identified that lacking rip
current knowledge was associated with rip current drownings, as was gender, age, alcohol
consumption, and overconfidence in swimming ability. Recent evidence suggests that while most
beach users are aware of rip currents and the hazard they pose, they are not able to identify a rip
current (Sherker et al., 2010; Caldwell et al., 2012; Brannstrom et al., 2014). More than 80% of
beach users surveyed in Florida and Texas failed to identify rip currents in photographs, usually
by incorrectly identifying areas of breaking waves as the most hazardous swimming conditions
(Brannstrom et al., 2014). This is consistent with results of Sherker et al. (2010) who argued that
most beach users are unable to identify a rip current and that "beachgoers clearly need to know
what a rip looks like to actively avoid swimming in it" (pg. 1787). Given sufficient information,
it is possible for beach users to identify a rip current with confidence (Hatfield et al., 2012).
However, the ability to identify a rip current or to recognize posted warnings about the rip
current danger is not a guarantee that a beach user will be safe, particularly because many will
still choose to swim in unsafe and unpatrolled sections of the beach, away from the presence of
lifeguards, for social or behavioral reasons or because of lack of awareness and/or complacency
(Drozdzewski et al. 2012; 2014; Williamson et al. 2012; Houser et al., 2016). Recent evidence
suggests that beach access management can inadvertently steer unsuspecting beach users towards
rip-prone areas, increasing the chances of a drowning occurring on that beach (see Barrett and
Houser, 2012; Houser et al., 2015; Trimble and Houser, 2017).

Informing the public about the rip current hazard has become a national priority in

several countries including the United States (e.g. Ashley and Black, 2008; Brannstrom et al.,
2013), Australia (e.g. Sherker et al., 2008; Brighton et al., 2013), United Kingdom (e.g.
Woodward et al., 2013), and Costa Rica (Aronzarena et al., 2015). The United States has
arguably the longest running cooperative and coordinated public rip current education program
operating across various organizational and political levels (Carey and Rogers, 2005). A Rip
Current Task Force was convened in 2003 by the National Oceanic Atmospheric Administration
(NOAA) and United States Lifesaving Association (USLA) to establish consistent rip current
education efforts and improve data sharing about rip current rescue data across the United States.
Subsequently, with the assistance of the National Weather Service (NWS) and Sea Grant, a
national "Break the Grip of the Rip!" ® education campaign was initiated in 2004. The "Break
the Grip of the Rip!" ® campaign aimed to educate the public about the rip current hazard by
providing information about what rip currents are, why they are dangerous, how to identify them,
what to do if caught in one, and how to help someone else if they are caught in a rip current.
Aspects of this information have been disseminated through various means such as the NWS Rip
Current Safety webpage (http://www.ripcurrents.noaa.gov/), brochures, beach signs, videos,
newspaper articles, and public service announcements on television.

While this campaign was the first of its kind globally, it was also particularly challenging given
that the United States has four very different coastlines (West Coast, East Coast, Gulf Coast,
Great Lakes) that differ in terms of wave climate and beach systems, and a large inland non-
coastal population who may only visit any of these coastlines infrequently. Results from
Brannstrom et al. (2015) suggest that while most beach users in Texas were not familiar with the
campaign itself, many were familiar with a key message of the campaign on "what to do" when
caught in a rip current.  This suggests that the campaign may have been successful in educating
beach users and reducing the number of drownings, but this hypothesis has never been formally
tested.

The core visual image used in many of these interventions was a simple diagrammatic

illustration of an idealized rip current from an oblique aerial perspective (Fig. 1).  In this image,
the rip current is characterized by relatively calm white water surrounded by more intensive
wave breaking adjacent to the rip and close to the shoreline. An image template was created that
could be accessed online and in hardcopy and duplicated freely to be posted along boardwalks,
beachfronts and public beach access points throughout the United States. The image has also
been more recently adopted in other countries such as Thailand, Costa Rica, Mexico, South
Korea, and Japan. While the NOAA-USLA sign was not intended to teach the general population
to identify a rip, the prominent image of a rip current on the sign and attempts to post the sign on
beaches indicate that its function and visual argument constitute an invitation to beach users to
use the information to identify rip currents (Brannstrom et al., 2015).

Due to this conflict between its' theoretical and practical use, the NOAA-USLA rip

current sign has proven to be mostly successful in regards to educating beachgoers on "what to
do" (e.g. swim parallel to the beach) when caught in a rip current, but has not been particularly
successful in improving beach users' ability to identify rip currents from the perspective of
standing or sitting on the beach (Brannstrom et al., 2015). Consistent with results of Matthews et
al. (2014), only a small percentage of beach users (<50%) recalled observing rip current warning
signs on beaches in Florida and Texas (Caldwell et al., 2012; Brannstrom et al., 2014) despite
their wide spread occurrence at beach access points. However, it is important to note that despite
observing and understanding a warning sign, it is well established that some people will not take
the appropriate actions to prepare for or avoid the hazard (Sietgrest and Gutscher, 2006; Karanci
et al., 2005; Hall and Slothower, 2009; Johannesdottir and Gisladottir, 2010).
In a separate initiative, the NWS has endeavored to develop a public rip current
forecasting system, although the methodology varies among Weather Forecast Offices (WFO).
Some WFOs issue surf zone forecasts that include a 3-tiered (low, moderate, high) rip current
outlook communicated to the public during television and radio news broadcasts (Carey and
Rogers, 2005) and social media platforms. Some WFOs work with local lifeguards to update
their outlooks based on real-time observations. However, as discussed in NOAA (2015), these
forecasts are not necessarily communicated or disseminated in a consistent manner throughout
all regions and therefore, are not communicated seamlessly. The lack of consistency in
forecasting is complicated by rip development being dependent on how the incident wave field
interacts with the pre-existing nearshore morphology, which is difficult to predict without local
knowledge on how it evolves over a range of spatial and temporal scales.
Since perception of the rip hazard depends in part on trust in experts and authorities, and
trust in the protective measures they employ (Njome et al., 2010; Heitz et al., 2009; Terpstra,
2009, 2011; Barnes, 2002), inaccuracies in the forecast or a discrepancy between the forecast and
what is observed at a specific beach at a specific time can erode confidence in the forecast
(Siegrist and Cvetkovich, 2000; Espluga et al., 2009).  Lack of confidence in forecasts could
potentially condition beach users to downplay the hazard warning on future visits (Hall and
Slothower, 2009; Scolobig et al., 2012; Green et al., 1991; Mileti and O'Brien, 1993).
Furthermore, the generic nature of the rip current forecasts can result in situations where the
actual intensity of rips varies substantially from the forecast. Beachgoers could easily observe a
discrepancy between their beach location and the rip forecast, caused by either the generalized
nature of the forecast or their inability to identify a rip current (Caldwell et al., 2012; Brannstrom
et al., 2014, 2015).
The national US rip current education program is clearly an impressive effort yet many
rip current related fatalities and rescues still occur on US beaches and overseas (Gensini and
Ashley 2010) and there is little quantitative evidence available to assess the overall effectiveness
of the program. This is largely because no 'pre-program' study was conducted on public
understanding, perception, or behavior in relation to the rip current hazard. There is also a lack of
hard data on rip current related fatalities, beach visitation numbers and how incident frequency
and exposure rate may have changed over time. In this regard, NOAA sponsored a workshop in
2015 to review the "Break the Grip of the Rip" ® program and NWS rip current forecasts to
discuss whether existing messaging is scientifically sound, as well as effective and clear in
reaching all age groups and demographics (NOAA, 2015).
It was acknowledged at the NOAA workshop that while there have been several recent
studies to describe the extent of rip current knowledge amongst beach users (or lack thereof) on
specific beaches in the United States (Caldwell et al., 2013; Brannstrom et al., 2014, 2015) there
is insufficient understanding about beach user knowledge of rip currents and their behavior at the
beach at a national level. This study describes results of a national online survey focused on
United States based beachgoers and their understanding of, and experience with, the "Break the
Grip of the Rip" ® program and the rip current hazard to provide quantitative evidence to guide
future improvements to beach safety education material and forecasting efforts.
**2 Methodology**
The study research design relied on an internet-based survey instrument using Qualtrics
approved by the relevant human subject protection program from Texas A&M University. The
survey consisted of questions re-phrased from Sherker et al. (2010) and photograph-based rip
current identification protocols (Fig. 2) modified from Brannstrom et al. (2014, 2015), with
questions grouped into six categories (Table 1). The survey had 75 questions and took
approximately 20-30 minutes to complete. It remained open from May-August 2015 and all
answers were recorded anonymously through Qualtrics Survey Software. A copy of the survey
instrument is provided as an appendix to this manuscript.
The survey was distributed by email to cooperating organizations and individuals for
distribution though listservs, websites, social media and in advertisements. In particular, it was
disseminated via secure Internet and social media links for Texas A&M University, Sea Grant,
Science of the Surf, NWS, and the National Oceanic and Atmospheric Association (NOAA).
While this internet-based recruitment process attempted to target a much wider demographic of
the US population, it is also reasonable to assume that as the host websites were all beach and
surf-related, survey respondents likely had greater interest in, and understanding of, coastal
environments and hazards leading to a potential bias that was also experienced in a beach safety
related study by Drozdzweski et al. (2012).

## 3 Results

Between May and August 2015, a total of 2084 respondents started the online survey, but only 1622 completed all questions (completion rate: 78%). Geographically, the largest number of respondents were from the state of Texas (n=368) where Texas Sea Grant and the local NWS office conducted significant advertisement for the survey. Large numbers of respondents also came from North Carolina (n=214), California (n=184), and Florida (n=130), with most remaining states having <50 respondents. Of the 50 US states, only Nebraska did not have a respondent. Overall this cohort managed to capture respondents who use each of the coastlines in the continental US. Respondents were evenly distributed by age (>18 years); each 10-year range between 21 and 60 garnered about between 320 and 420 respondents. A slight majority of the respondents were female (55%).

### 3.1 Familiarity with the Break the Grip of the Rip ® Campaign

Only 18% (n=304) of respondents reported hearing about the Break the Grip of the Rip ® Campaign with nearly identical split by gender and age. Approximately 40% of respondents reported hearing about the campaign either through a brochure/pamphlet (n=120) or at the entrance to a beach (n=119), whereas 163 respondents (54%) reported hearing about the campaign through various sources on the internet. 90 respondents reported having heard about the campaign from the Break the Grip of the Rip ® website. When asked what Break the Grip of the Rip means, most respondents (familiar with the campaign) reported (to varying degrees of accuracy) that it was designed to provide information about what to do if caught in a rip current:

*Do not try to fight the current, instead work with the current until you can break free of its pull*

*Advises affected swimmers not to struggle while heading shoreward*

*but to swim parallel to the beach till out of the off-beach current*

There were, however, several respondents (familiar with the campaign) who believed that the

messaging was not appropriate and needed to be rethought:

*The slogan is useless to anyone caught in a rip current!*
*What can you do by knowing this slogan? ...."Wave, Yell & Swim Parallel"*
*is a far better slogan...it provides 3 lifesaving pieces of information. The existing slogan*
*provides nothing.*
*it's an advertising slogan; it doesn't mean much at all.*
*It's a bad slogan; it does not tell folks what to do,*
*what to watch for, or anything useful.*

Responses from those who were not familiar with the campaign were much shorter and did not
contain the level about survival strategies provided by those familiar with the campaign.
Representative responses include "how to escape", "tips to survive", and "how to get out of a
rip".

**3.2  Beach Preference**

As presented in Fig. 3, most respondents visited the beach either once per year on

vacation (22%) or multiple times per year (42%). Visitation exhibits a statistically significant
relationship with age, with older respondents (>40) visiting the beach more often than younger
respondents ($\chi^2$=46.5, $\rho$<0.01). Perceived wave size on beaches visited by respondents depends
on age and frequency of beach visitation with older respondents who visit the beach frequently
tending to report beaches they visited having strong waves, while younger respondents, who
tended to visit the beach infrequently, identified the beach as having small waves ($\chi^2$=84,
$\rho$<0.01). In general, respondents who visit the beach infrequently tend to describe the beach as
having small waves and that their primary beach activity is swimming and/or wading. All
respondents who visit the beach frequently (weekly or daily) identified board riding as their main
activity and tended to frequent beaches with strong wave activity ($\chi^2$=111, $\rho$<0.01), suggesting a
greater understanding of wave conditions. There was no statistically significant variation in wave
description based on home state, suggesting that perception of wave activity is largely based on
frequency of beach visitation and other personal characteristics.  In terms of choice of beach
visited, wave activity and the potential hazard posed by rip currents or the absence of lifeguards
is less important than cleanliness and at the same level of importance as crowds (Fig. 4).
When determining which beach to visit, frequent beach users, who were mostly board
riders, tended to prefer beaches with lots of waves, whereas infrequent users emphasized safety
and cleanliness ($\chi^2$=159, $\rho$<0.01). Frequent beach users also believed it was very important to
swim near a lifeguard, while infrequent users did not ($\chi^2$=51, $\rho$<0.01). Across both groups,
however, respondents suggested they would still enter the water even if a lifeguard was not
present, suggesting that recognition about the importance of lifeguards is not consistent with
behavior in selecting where and when to swim (Fig. 5).   Frequent beach visitors were also more
confident in their ability to 'always' spot a rip current in contrast to infrequent beach visitors
($\chi^2$=247, $\rho$<0.01). Those who visit the beach less often (e.g. several times per year or month)
believed they could spot a rip 'sometimes' or believed it is not possible to see a rip current,
consistent with the response from all respondents (Fig. 6).
**3.3 Swimming Ability**
Most respondents (~52%) self-identified as competent swimmers (Fig. 7) and reported in
a separate question that they were capable of swimming between 25 and 100 yards (or more than
100 yards) without having to stop or pause in open water ($\chi^2$=1391, $\rho$<0.01). Respondents who
self-reported as *highly* competent open water swimmers (n=213, 12%) primarily believed they
could swim more than 500 yards in open water without resting, while those who self-reported as
weak swimmers (n=566, 31%) believed that they were only capable of swimming 25 yards or
less. Those who identified as highly competent or weak swimmers tended to have the narrowest
range of self-reported ranges of swimming ability, while those who self-identified as competent
swimmers had the widest range of self-reported swimming distances for both pools and open
water.
**3.4 Ability to Identify a Rip Current**
When asked "Where on this photograph would you swim?", approximately 54% of
respondents correctly identified the location furthest away from the rip current in Photograph 1
(Figs. 2a and 8a). However, 182 (11%) respondents incorrectly selected the rip current as the
safest location to enter the water, with the remaining respondents identifying other areas of the
photograph (adjacent to the rip) as being the safest location. Results of a z-test suggest that
respondents who selected the rip as the safest location are significantly younger than those who
correctly identified the safest location in the photograph ($z=12.1$, $\rho<0.01$). Those who correctly
identified the safest location in the photograph also visited beaches more frequently ($z=6.1$,
$\rho<0.01$) and self-reported beaches they visited as having strong waves ($z=6.4$, $\rho<0.01$). Most
respondents who identified the rip as the safest location self-reported never having swimming
lessons ($z=2.8$, $\rho<0.01$) and described themselves as weak swimmers in both pools ($z=3.7$,
$\rho<0.01$) and open water ($z=6.2$, $\rho<0.01$). Those same respondents also self-reported that it was
important to swim near a lifeguard ($z=5.8$, $\rho<0.01$), but tended to not consider hazards before
going to the beach, unlike respondents who were able to correctly identify the safest spot to enter
the water ($z=14.1$, $\rho<0.01$).
When asked what beach features they believed to be most dangerous, respondents who
correctly identified the safest swimming location away from the rip were more likely to report
alongshore currents and rip currents as dangerous features, while those who selected the rip as
the safest location tended to identify jellyfish, sharks, and big waves. Respondents who
incorrectly selected the rip current as the safest location were also least familiar with the
common US beach safety flag system ($z=11.5$, $\rho<0.01$), and tended to have not heard of rip
currents ($z=17.3$, $\rho<0.01$). Respondents who selected the rip as the safest location did not
understand what was meant by a "high risk" ($z=3.2$, $\rho<0.01$) or a "low risk" ($z=7.5$, $\rho<0.01$) of
rip current development as broadcast by some NWS services. The same respondents also noted
that rip forecasts are apt to be inconsistent with the conditions they encountered on the beach, in
contrast to respondents who correctly identified the safest location in the photograph and noted
that forecasts tended to be consistent with their experience ($z=3.3$, $\rho<0.01$).
Approximately 25% of respondents (n=630) incorrectly identified the left side of the
groin (with an active rip) as the safest spot to enter the water in Photograph 2 (Figs. 2b and 8b).
Like the responses to Photograph 1, those respondents tended to be younger ($z=5.2$, $\rho<0.01$), go
to the beach infrequently ($z=7.8$, $\rho<0.01$), and self-report waves being relatively small ($z=7.3$,
$\rho<0.01$) and their swimming ability in open water to be relatively poor ($z=2.2$, $\rho<0.01$). These
respondents are also unlikely to consider hazards before going to the beach ($z=10.9$, $\rho<0.01$), are
unfamiliar with the common beach flag system in the United States ($z=12.5$, $\rho<0.01$), do not
understand the definition of a "high-risk" of rip current development ($z=4.2$, $\rho<0.01$), and
believe that rip forecasts are not consistent with their personal beach experiences ($z=2.8$,
$\rho<0.01$). Unlike responses for Photograph 1, those respondents who incorrectly identified the rip
as the safest location were not significantly different (at the 95% confidence level) from those
who correctly identified the safest location (right side of the groin) with respect to: pool
swimming, swimming near a lifeguard, type of water activity at the beach, knowledge of the
"*Break the Grip of the Rip*" ® campaign, or their perceived ability to use the sign to identify a rip
current.
A similar pattern was observed in respondent' ability to identify the safest location to
enter the water in Photograph 3 (Figs. 2c and 8c), with 26% of respondents incorrectly
identifying the rip current as the safest location.  Like responses for the other photographs,
respondents who identified the rip as the safest location to enter the water did not visit beaches as
often ($z=4.5$, $\rho<0.01$), self-reported having relatively limited swimming ability in pools ($z=3.1$,
$\rho<0.01$) and open water ($z=2.8$, $\rho<0.01$), and did not believe it was important to swim near a
lifeguard ($z=3.0$, $\rho<0.01$), unlike those who correctly identified the safest location to enter the
water in the photograph. Respondents who selected the rip current as safe for swimming were
not as familiar with the flag system used in the United States ($z=5.6$, $\rho<0.01$), rip currents ($z=3.9$,
$\rho<0.01$), or the "*Break the Grip of the Rip*" ® campaign ($z=4.4$, $\rho<0.01$).  These respondents also
did not understand what was meant by a "low risk" ($z=2.5$, $\rho<0.01$) and a "high risk" ($z=3.4$,
$\rho<0.01$) of rips. However, unlike Photographs 1 and 2, no statistically significant difference was
observed between those who correctly or incorrectly identified the safest spot to enter the water
with respect to: age, self-reported wave activity, swimming lessons, behavior in the absence of
lifeguards, importance of checking for hazards, or the ability to use the sign to identify a rip
current.
**3.5  Response to the Rip Current Warning Sign**
Only 31% of all respondents believed the NOAA rip current warning sign could be used
to identify a rip current. Interestingly, those respondents who incorrectly identified the rip
current as the safest spot on the beach to enter the water tended to believe that the NOAA rip
current warning sign could *not* help a beach user identify a rip current. This contrasted with those
who correctly identified the safest location in any of the photographs ($z=5.2$, $\rho<0.01$). When
asked to describe how the sign could be used to identify a rip current, some of the latter
respondents were able to relate the rip in the picture to a real rip:
*It shows that in a rip current, there appears to be a break in the water, with water*
*moving in a different direction.*
*It shows you the "calm" area between the two areas of normal wave activity*
*indicating the channel where the rip is located*
Most of these responses focused on the pattern of wave breaking and the orientation of the
'calmer' water to the beach. There is evidence that some respondents believed the picture to be
an accurate representation of a rip, but they could not provide specific detail about the real-world
features on the beach it depicted, for example "*Graphic depiction of what the tide looks like.*"
This suggests that some respondents believe the sign is accurate since it was designed and placed
there by an authority.
As previously noted, the rip current warning sign was not designed to help beach users
identify a rip current, but rather to inform them how to escape a rip. Most respondents could
clearly state what the sign was informing them about swimming parallel to the beach to escape a
rip:

*Let the current take you out and then swim parallel the shore to escape.*
*Swim parallel to the shore, or wait until the rip gets less strong further offshore.*

96% of respondents could provide a response to this question and virtually all responses
indicated that the sign informed them to swim parallel to shore to escape the rip current,
suggesting that the sign has been effective in communicating this message. When asked how
seeing this sign would change their behavior of the beach, a majority (65%) of respondents
suggested they would take precaution when entering the water:
*Might avoid going in water if I see surface signs of rip activity and drive to*
*another beach*
*Consider not going in. Look carefully for signs of rips. Look for flags and*
*lifeguards*
This suggests that while most respondents understood that the sign provided them with
information on how to escape a rip current, it also helped with prevention as most respondents
also noted that they would take precaution or use it to spot (and presumably avoid) a rip, rather
than focus on escape strategies.

Most respondents (86%) provided ideas on how to improve the rip current warning sign,

with more than half suggesting the sign needed to provide a more accurate depiction and/or
description of a rip current:
*I don't think it clearly identifies it enough that the waves will not break where a*
*rip current is. It is great because it shows how to get out of one but I think with*
*another picture of an actual rip current people would identify them easier.*
*Pictures showing what actual rip currents look like would be useful. / Most casual*
*beachgoers are not confident that they could identify a rip current from shore or*
*predict where one might be forming.*
*There needs to be more info on how to detect, recognize and avoid a rip current.*
*Information on conditions during which rip currents are most likely to form would*
*also be useful.*
A small number of respondents (<10%) suggested that the sign should either include step-by-step
instructions on what to do and/or provide more information about the experience of being caught
in a rip current:

*Multiple steps: / 1. Know when you're in a rip / 2. Stay calm and tread water / 3. Wait until you've floated out to a slower moving water. / 4. Swim sideways*

*Specific instructions on what one should do if caught in a rip current - Should I swim left, right, straight? What if I'm not a strong swimmer? What are some other exit options?*

Another group of respondents (~15%) either did not provide suggestions on how the sign can be improved or noted that it only needed minor edits, including space for local emergency numbers and contacts. A small number of respondents (<5%) believed that the sign should include statements that elicit fear amongst beach users including statements such as "Rip currents can drown you."

## 3.6 Prevention

One in four (25%) respondents reported they had been previously caught in a rip current by accident, while 10% of respondents reported that they had purposely entered a rip for surfing. When asked how to escape a rip, those who had accidently been caught in a rip current provided relatively detailed responses that either described escape by swimming parallel or riding the current without panic:

*Let it flow. Don't fight it.  Perhaps as long as you minimize tiring exertions try to flow towards the side of the current. Basically do the same thing you'd do if you fell in a strong river about to empty into a lake. You certainly wouldn't kill yourself trying to swim out upstream.*

*Don't panic!!!  Either swim - without too much exertion - parallel to the beach for 25+ yards, OR tread water and allow yourself to be carried out until the rip loses power, then swim parallel to the beach. Once out of the rip, swim back towards shore (again in a relaxed manner, taking time to prevent exhaustion). When nearing the beach, take care not to get drawn back into the rip by water flow parallel to the shoreline.*

Of those who had not been previously been in a rip 7% (n=36) did not provide a description of
how to escape. The remaining respondents provided relatively short responses that described
escape through combinations of swimming parallel and relaxation
Assuming no response is an indication of a lack of knowledge about rips, the number of
respondents who did not provide an accurate description of how to escape a rip current is ~9%,
suggesting that overall the campaign has been successful in informing beach users to: 1) not fight
the current; 2) swim out of the current, then to shore; 3) if you can't escape, float or tread water;
and 4) if you need help, call or wave for assistance.
**3.7 Forecasts**
Respondents were also asked about whether they were aware of rip forecasts, if forecasts
altered their behavior, and if the forecasts conformed with their observations at the beach. Since
existing rip forecasts are not consistent and few are based on an understanding of pre-existing
morphology, the focus here was not on the actual accuracy of the forecast, but on whether the
respondent believed the message to be consistent with their observations. About half of
respondents (52%) reported seeking information about beach and surf conditions before going to
a beach with the majority (83%) using the internet to find that information. A large majority
(88%) of respondents stated that information about beach and surf conditions affected their
behavior, with many saying that they would either "not go" (to the beach), "not go in the water",
or "look for rips". When asked whether the rip current forecast (either high or low) was
consistent with conditions they experienced at the beach, approximately 67% of respondents
stated that the forecasts were not necessarily consistent with their observations. For some, this
inconsistency reflected the temporal and spatial broadness of the rip forecast compared to what
they observed:
*Weather changed quickly and no beach flags were posted, advising of rip*
*currents.*
*Rip currents cannot be predicted for individual beaches, they are blanket*
*warnings.*
Other respondents noted the forecast was inaccurate because other beach users had not adjusted
their behavior:
*I never noticed an[y] thing unusual and people in general don't seem to adjust*
*their behavior.*
Others noted it was not possible to determine if the forecast was accurate because they were not
able to spot a rip on the beach at that specific time or in general:
*I couldn't determine if/where rip tide activity might be in the water if the forecasts*
*had warned beach-goers to be aware of a high risk on that day.*
In several cases (n=59), respondents noted they had not heard a forecast warning of the rip
hazard on a given day or in general through responses such as "*I don't know if I've ever heard a*
*rip current forecast?*"

Additional questions about high-risk rip conditions solicited written responses that

suggest many respondents understood the high-risk warning to mean that wind and wave activity
are tantamount to the development of rips:

*Due to tides, weather, etc., there is a much greater risk for rip currents in the*
*ocean.*
There was a mix of responses in which respondents believed that 'high risk' meant that rips
would form or that there was a greater chance of rip formation. Others (n=102) believed that the
use of the terms high and low risk were misleading:
*Whenever or wherever there are waves there can be rip currents, so I am not sure*
*what 'high' or 'low' risk of rip currents means. All rips are potentially*
*dangerous.*
In response to the definition of low risk, respondents tended to suggest this implied that rips were
unlikely or would not form:
*Rip currents may still exist but are weaker or fewer than normal.*
*Conditions are not conducive to rip currents.*
*The factors necessary for rip currents to form are absent- not likely to encounter rip.*
Of note, whether a respondent described high and low risk of rips as a probability (likely,
unlikely) or in absolute terms (is or is not present) is not related to whether the respondent noted
that the rip forecast was consistent with their observations at the beach. For both high and low-
risk, some respondents believed that the forecast (by radio, internet, etc.) was not based on the
predicted weather, but rather on whether a rip had been sited on a beach or not with statements
such as: *"Not Sighted"* or *"Strong rips observed."* Others (n=129) believed that high and low
risk was associated with the local bathymetry being conducive to the formation of rips: *"the*
*topography/bathymetry is suited to rip currents."*
**3.8 Trusted Sources of Information**

Respondents were also asked to rank sources of information about rip currents from (1)

most trusted to (5) least trusted. Except for social media (including Facebook, Twitter, etc.), all
sources of information were nearly equally ranked from most to least trusted with no discernable
pattern. Only social media exhibited a discernable pattern, with more than 35% of respondents
identifying it as the least trusted source, although 18% of respondents also identified it as the
most trusted. More respondents identified internet sources as the most trusted compared to other
sources, while television and radio were identified as trusted (rank 2 and 3), but not the most
trusted.  No significant correlations were observed between trust in a source of information and
respondent demographics, suggesting that a broad communication strategy is the most effective
to reach the widest audience.
**4 Discussion**
The primary results of this US-based rip current survey are summarised in Table 2.

Results suggest that while many are not aware of the "Break the Grip of the Rip" ® campaign,
the US beach-going public is informed about rip current safety. While this is an encouraging
result, it needs to be placed in context. The goal of this study was to examine United States based
beachgoers understanding of, and experience with, the national "Break the Grip of the Rip" ®
program and the rip current hazard to provide quantitative evidence for improving the program.
Despite the dissemination of the online survey leading to a potentially biased cohort (Section 2)
that was dominated by respondents who were relatively frequent beachgoers, self-rated as
competent swimmers, and were able to successfully identify the safest location to enter the water
based on photographs, approximately 10% of survey respondents were infrequent beachgoers,
poor swimmers and largely ignorant of the rip current hazard and more liable to make poor swim
location choices.

When taking the entire US beachgoing population into account, this cohort represents a
significant population of potential 'at risk' beachgoers. Given that this population was a key
target of the "Break the Grip of the Rip" ® campaign, it is therefore of considerable concern that
this cohort: i) tended to select the rip current as the safest location to enter the water on each of
the survey photographs; ii) did not consider hazards before going to the beach; iii) were not
familiar with the beach flag system in the United States;  and iv) did not seek out lifeguards
when visiting a beach. These results clearly highlight how at risk infrequent beach users still are
despite the decadal existence and ongoing presence of the campaign.
In contrast, survey respondents who were frequent beachgoers and had previous
experience with rip currents had  a better understanding of  what rip currents were, the danger
they represent and how to escape from a rip. As described by Brannstrom and Houser (2015),
those who get caught in a rip current "*understand the dangers of rips first hand and…. realize*
*[they] never want to be caught in that situation or accident [again].*" Similar results were found
in studies involving surveys of people who had been caught in rip currents in Australia
(Drozdzewski et al., 2012; 2015). Those with indirect or no experience tend to underestimate the
danger compared to those with direct experience (Ruin et al., 2007).
It is also interesting to note that while many survey respondents were not familiar with
the "*Break the Grip of the Rip*" ® campaign itself, a clear majority (~91%) understood  the
primary message of the campaign and were able to provide an accurate explanation of the
message (i.e. "break the grip"). Respondents previously familiar with the campaign provided
detailed explanations of how to escape a rip by swimming parallel and/or floating until the
current weakened, indicating they may also have gained this knowledge from other sources.

Survey results also suggest that other factors can influence behavioral response in relation

to the rip current hazard. For example, as noted by several survey respondents, if everyone else
at the beach is entering the water and not heeding an existing rip current warning (out of
ignorance or purposeful neglect) there is a chance that the beach user may become complacent
and also enter the water despite understanding the risk. This suggests that decisions can be made
based on what other beach users are doing rather than rip forecasts (Lapinski et al., 2014). The
tendency to follow the behavior of others may be enhanced when someone goes together as part
of a group and enters the water because everyone is willfully ignoring the risk or is ignorant to
the severity of the risk (see Mollen et al., 2012; Aronzarena et al., 2015). A regional forecast or
global warning will not necessarily deter beach user behavior as much as direct intervention by
lifeguards.

This study has also revealed some important issues with existing rip forecasting methods

and resultant warnings (Table 2). Approximately 67% of all respondents stated that rip current
forecasts are not necessarily consistent with what they observe on the beach. Consistent with
previous studies on natural hazards, those who have not experienced a predicted hazard or did
not experience personal damage during a visit to the beach are more likely to downplay the
danger the next time they visit (Hall and Slothower, 2009; Scolobig et al., 2012; Green et al.,
1991; Mileti and O'Brien, 1993). Any inconsistency between a rip forecast and direct
observations therefore has the potential for some beach users to downplay the rip current risk on
future beach visits.. While forecast methodology varies by WFO, most rip forecasts do not
consider bathymetry, local topography, or hard structures that may force rips over a range of
wind wave conditions. It is also not clear how many forecasts are based on the actual presence of
rips observed by lifeguards.
The key problem is that rip forecasts tend to be generalized for a large region and time,
whereas actual rip development and flow behavior is extremely variable over space and time
(Castelle et al., 2016).  It is also difficult to predict the potential for rip development without an
understanding of the pre-existing nearshore morphology, which itself is difficult to measure
directly, remotely or through numerical modelling. A static daily regional rip warning may
therefore fail to replicate different rip conditions that occur during that day For beachgoers, this
can lead to a different interpretation of the forecast accuracy and may potentially lead to
downplaying the actual risk (see Brilly and Polic, 2005). Mileti and O'Brien (1993, p 40)
describe this reasoning as "*The first impact did not affect me negatively, therefore, subsequent*
*impacts will also avoid me."* At the same time, beach users will not be able to conceptualize
events that have never occurred or to see future trips to the beach as anything more than a mirror
of past visits or experiences (Kates, 1962; Tversky and Kahneman, 1973).  If the rip forecast and
warnings are inaccurate or perceived to be inaccurate by the beach user, there may also be a
potential loss of trust in that authority (Espluga et al., 2009) and future forecasts.
It can be assumed that beach users who rely heavily on rip forecasts and assume they are
accurate might use them to calibrate their own observations and experiences, which will impact
their future forecast expectations. If a low rip risk forecast is issued and the rips are actually
prevalent and strong, then beach users may lose faith in forecast accuracy. Similarly, if a high rip
risk forecast is issued and no rips are observed with relatively calm conditions, then beach users
may become complacent about the hazard and discount or ignore future forecasts in the future.
However, results of this study suggest that given time and experience at the beach over a range
of conditions, beach users can develop a nuanced understanding of the forecast and gain greater
confidence that it is appropriate.  Rip forecast inaccuracies appear to be most problematic for
infrequent beach users who also do not appear to seek out lifeguards and are unable to spot rips
correctly.
A majority of respondents were able to clearly state what the standardized rip current sign
was informing them to do in terms of swimming parallel to the beach to escape a the rip, but
many identified a need to provide information that would allow beach users to identify a rip
current in general (e.g. "*Pictures showing what actual rip currents look like would be useful*") or
specific to the local beach (e.g. "*Picture of rip at actual beach [the sign] is placed on*").
However, evidence from beach surveys in Florida and Texas suggest that beach users are not
able to accurately identify a rip current (Caldwell et al., 2012; Brannstrom et al., 2014), although
there may be ways in which the sign can be made more accurate through small revisions to the
perspective, colors, and beach morphology (Brannstrom et al., 2015).  While local information
may improve the accuracy and interpretation of the sign, there is the potential for different signs
and messaging being used (of varying quality and detail), leading to confusion and
misinterpretation by beach users.  A more appropriate strategy may be to take a more local-
approach to risk and emergency management including local emergency contact information.
This approach places greater authority in local managers and emergency responders, without
resulting in different signs.
A local approach also includes putting greater emphasis on the expertise of lifeguards to
prevent accidents and respond to emergencies promptly and properly. This would also partially
consider the fact that there are different types of rip currents and associated behavior in different
geographic locations and regions (Castelle et al., 2016). Of note, Surf Life Saving Australia has
recently adopted a 'combined approach' to promoting how to escape a rip current (Bradstreet et
al., 2014). This decision was largely based on field tests of rip escape strategies (McCarroll et al.,
2014; Van Leeuwen et al., 2016), which clearly showed that natural variance in rip flow behavior
influences effectiveness of different rip escape strategy strategies. This has also been illustrated
by recent numerical modelling studies (McCarroll et al., 2016; Castelle et al., 2016). However,
communicating such a complex and mixed message is problematic. In contrast, concepts of rip
avoidance instruction are consistent and simpler to explain, making them more suitable for
advertising campaigns and signage (Bradstreet et al. 2014).

While there is still insufficient evidence to suggest that present warning systems help

people avoid and escape rip currents (see also Lapinski et al., 2014), there is evidence that
lifeguards are effective at preventing drowning death through preventive actions and rescues.
With proper training and experience a lifeguard can provide invaluable local understanding of
the rip hazard to provide effective mitigation. Unfortunately, there is no consensus amongst
beach users that it is safe (or not) to swim in the surf after lifeguards are off duty (Petrass and
Blitvich, 2014), despite evidence that it is safer to swim in the presence of a lifeguard. In this
respect, greater focus should be placed on reminding beach users to swim near lifeguards and
only at times that lifeguards are present because "the chances of drowning at a beach protected
by lifeguards trained under USLA standards is less than one in 18 million" (Branche et al. 2001).
**5 Conclusions**

A survey about the extent of public rip current knowledge in the United States was

conducted with the aim of establishing a dataset that provides guidance for the improvement and
enhancement of existing beach safety interventions. Results suggest that the US-based "Break
the Grip of the Rip" ® campaign has been successful in helping inform the public about rip
current safety. Although few respondents were familiar with the  campaign itself, most
respondents could provide an accurate description of how to escape a rip current by swimming
parallel and/or floating until the current weakened.  Results suggest that the most at-risk
population are infrequent beach users because they do not seek out lifeguards, do not take the
same precautions as frequent beach users, and believe there are large discrepancies between rip
forecasts and their own observations at the beach.  Survey results provide a conservative estimate
of 10% of US beachgoers being at risk of being caught in a rip due to ignorance and/or poor
swimming choices.   Future education efforts should attempt to target this beachgoing
demographic group. Knowledge of rips, visual ability to accurately identify a safe swimming
location in where rip currents are present, and ability to interpret rip forecasts are each dependent
on prior experience with rips and the frequency of beach visitation.  In addition to concerns
about the spatial and temporal accuracy of public rip forecasts, many respondents identified a
lack of local detail in the rip current warning sign as a concern, with more than half of
respondents suggesting the sign needed to provide a more accurate depiction and/or description
of a rip current and local emergency information.  This suggests a need for greater focus on
locally specific and verified rip forecasts and signage in coordination with lifeguards, but not at
the expense of the successful "Break the Grip of the Rip" ® campaign.

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

**Tables**
**Table 1.** Question groups used to elicit responses from respondents notified about the survey by
various agencies in the United States.

| Group | Focus of Questions | Example topics |
|---|---|---|
| 1 | Informed Consent | |
| 2 | Non-identifying personal information | ZIP code, age, ethnicity, and beach use |
| 3 | Swimming behavior | Self-assessed swimming ability |
| 4 | Beach behavior and beach safety information | Frequency of visits; perceived risks at the beach |
| 5 | Rip identification and knowledge | Description of a rip current; ability to identify rip current in a photograph |
| 6 | Memorability, conspicuity, comprehension, priming | Source of rip information; memory of observing rip safety warnings |
| 7 | Rip current sign knowledge and understanding | Understanding rip current warning sign and warnings |


**Table 2.** Summary of major findings from the "Break the Grip of the Rip!" ® National Rip
Current Survey.

| Focus of Questions | Example topics |
|---|---|
| Beach Preference | • Frequency and purpose of visits to a beach affect perception of surf conditions, importance of swimming near a lifeguard and self-reported ability to spot a rip current |
| Swimming Ability | • Range of self-reported swimming ability (distance in open water) related to self-reported competency |
| Ability to Identify a Rip Current | • Ability to identify safest location in a photograph related to frequency of beach visits, self-reported swimming competency and training<br>• Ability to identify safest location related to perceived importance of and concern about surf hazards, self-reported understanding of "high" and "low" risk conditions, and perceived accuracy of rip forecasts |
| Response to Warning Sign | • Perceived ability to use sign to identify a rip current varied with ability to identify safest location on a photograph<br>• Sign has been effective in communicating swimming parallel as an escape strategy, and taking caution when entering the water<br>• Identified need to provide a more accurate depiction of a rip current, detailed instructions on how to escape a rip current, and local emergency information |
| Prevention | • "Break the Grip of the Rip" ® Campaign has been successful in informing beach users to: 1) not fight the current, 2) swim out of the current, then to shore, 3) if you can't escape, float or tread water, and 4) if you need help, call or wave for assistance |
| Forecasts | • Self-reported change in behavior based on forecasted beach and surf conditions, but tendency for forecasts to be inconsistent with observations<br>• Perceived inaccuracy of forecast related to spatial and temporal broadness of forecast, inability to identify a rip, and behavior of other beach users |
| Trusted Sources of Information | • No significant correlations were observed between trust in a source of information and respondent demographics |

**Figures**

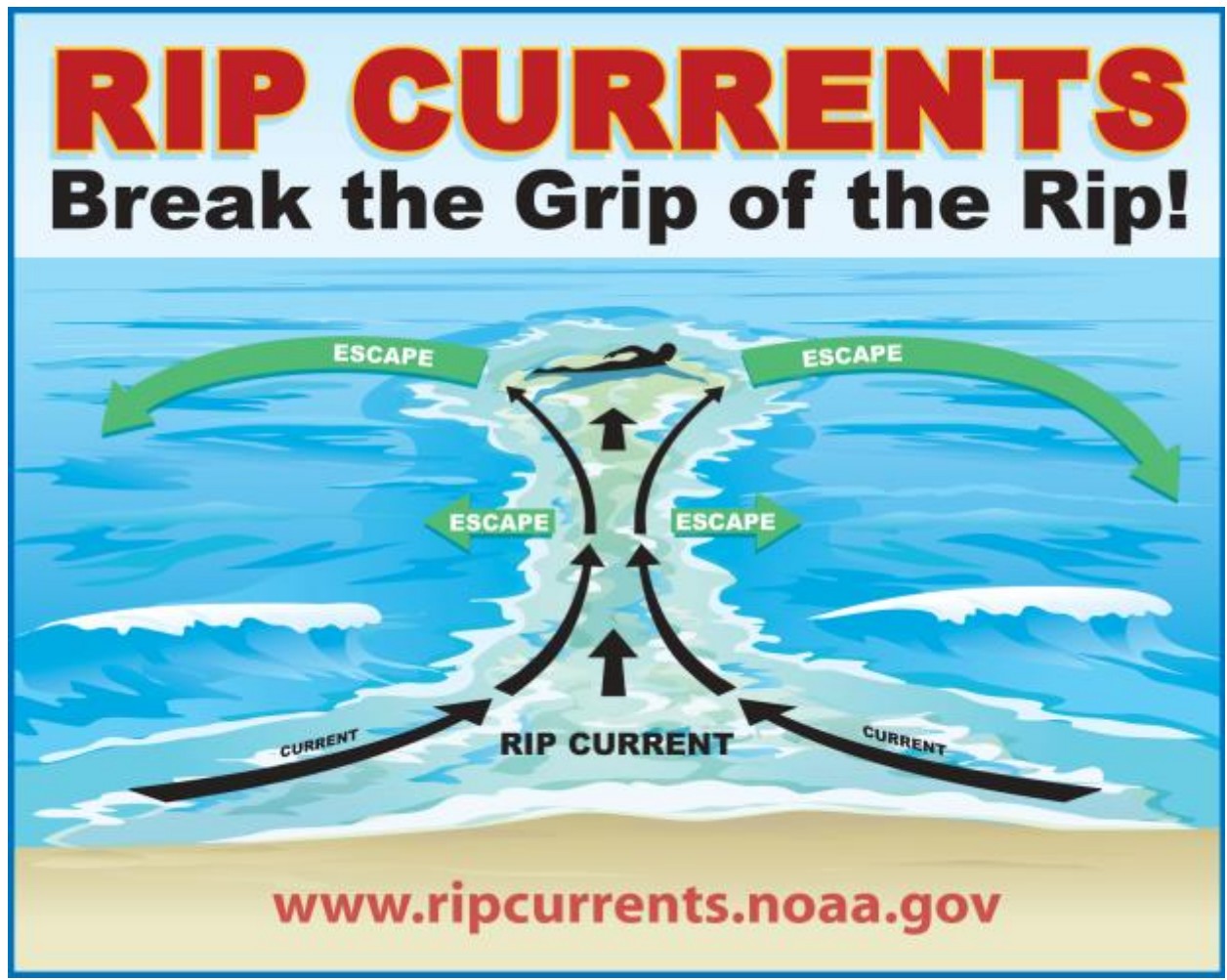

**Fig. 1.** Rip current warning sign developed by the United States Rip Current Task Force as part
of the "Break the Grip of the Rip!" ® education campaign.

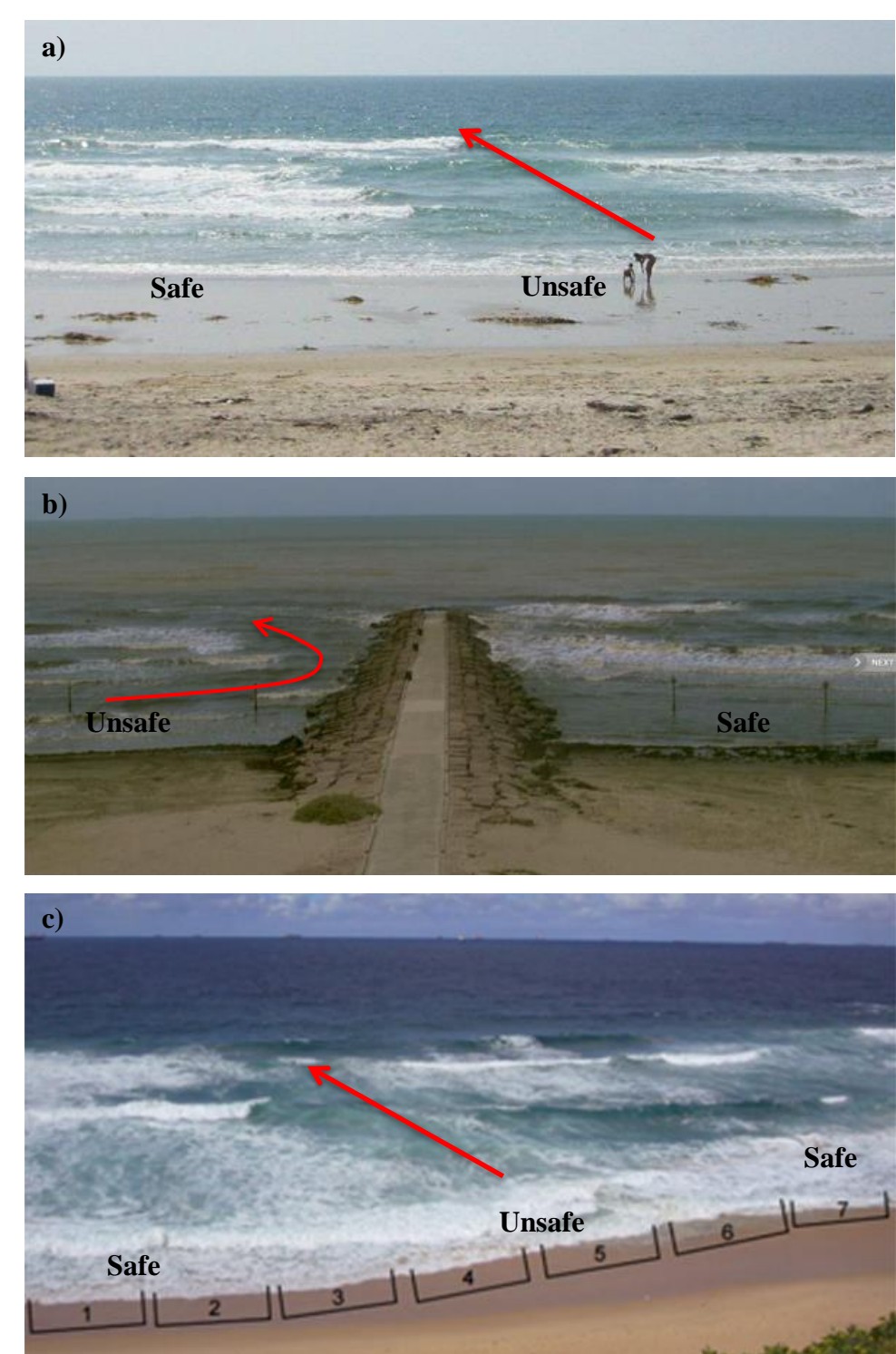

**Fig. 2.** Photographs used in Questions 42 through 44 of the survey to ask respondents "Where on this photograph would you swim?". The location of the rip current in each photograph is shown by the red arrow, which was not visible to the respondents.

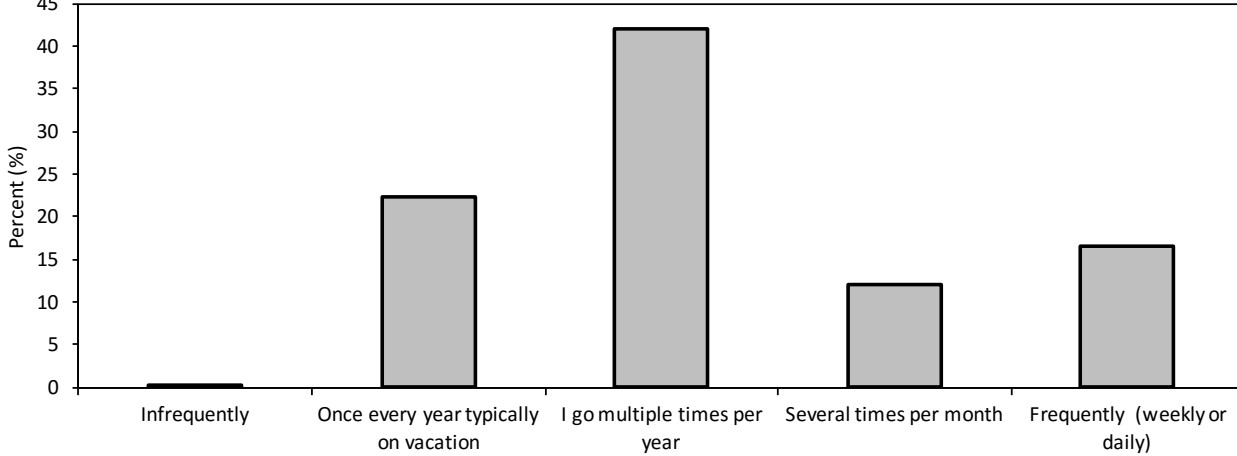

**Fig. 3.** Percent of self-reported beach visitation by respondents.

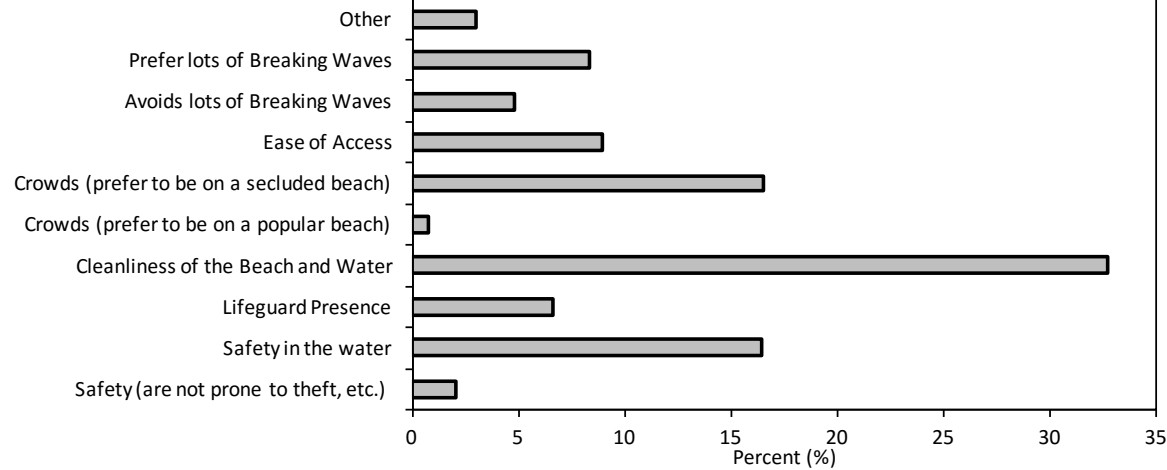

**Fig. 4.** Relative importance of beach and surf factors to respondents when selecting a beach. Note that respondents were asked to identify all factors that applied.

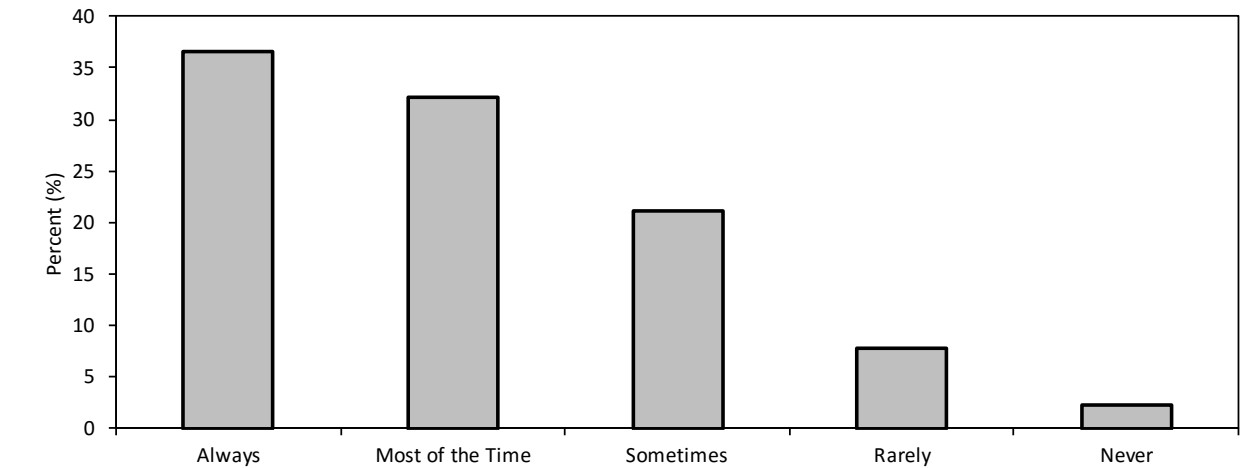

Fig. 5. Self-reported tendency to enter the water in the absence of a lifeguard on a beach.

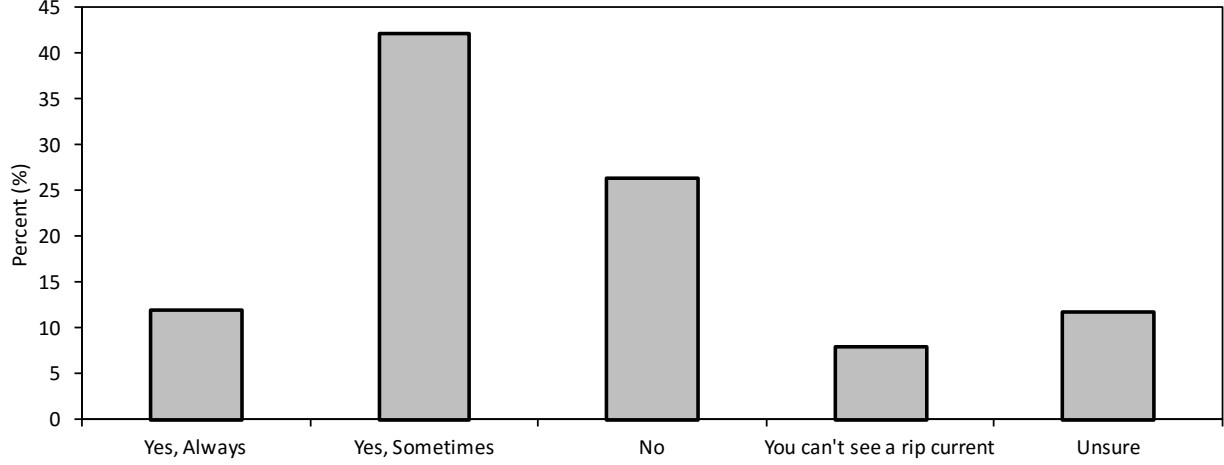

Fig. 6. Percent of respondents' belief that rip currents can be seen by beach users.


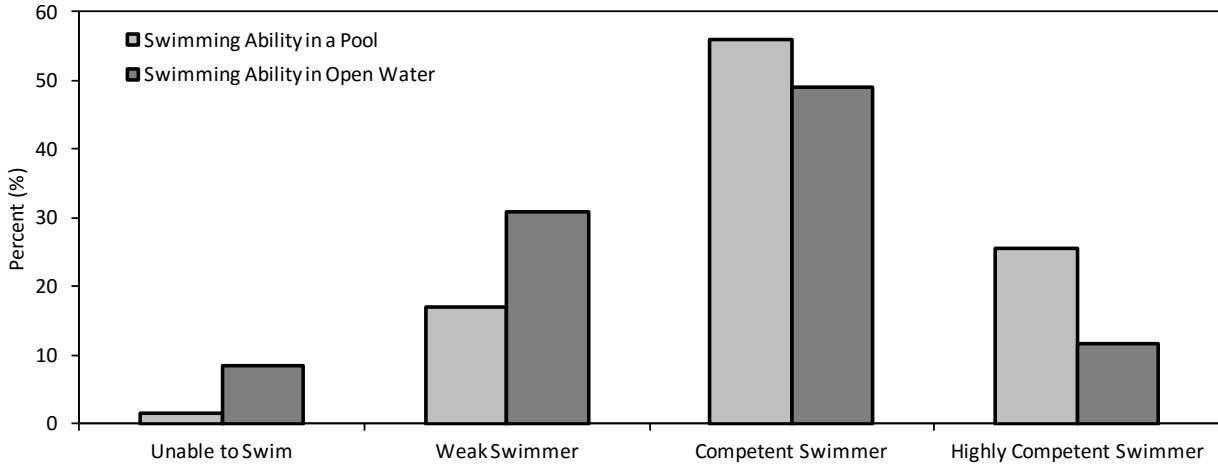

**Fig. 7.** Percent of self-reported swimming ability in a pool and in open water with waves.

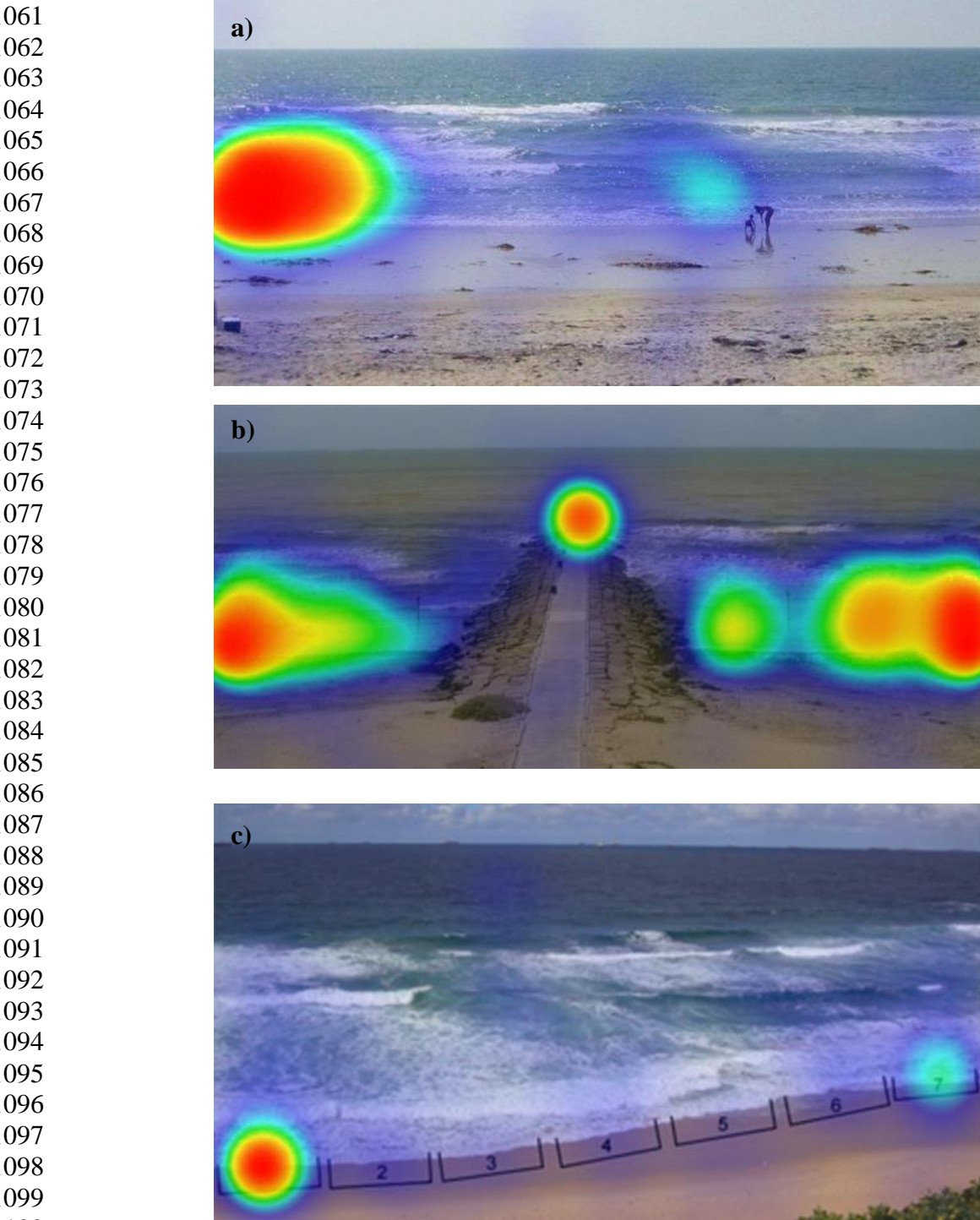

**Fig. 8.** Identified location of safest location to enter the water in the photographs presented in Question 42 through 44 and also presented in Figure 2. Warm (red) colors indicate large number of responses, while cold (blue) colors indicate few responses. No color (background picture) represents areas that received no responses.

**Appendix 1**
Q1 Are you a resident of the United States?
❍ Yes (1)
❍ No (2)
Answer If Are you a resident of the United States? Yes Is Selected
Q2 In which state do you currently reside?
❍ Alabama (1)
❍ Alaska (2)
❍ Arizona (3)
❍ Arkansas (4)
❍ California (5)
❍ Colorado (6)
❍ Connecticut (7)
❍ Delaware
❍ (8)
❍ District of Columbia (9)
❍ Florida (10)
❍ Georgia (11)
❍ Hawaii (12)
❍ Idaho (13)
❍ Illinois (14)
❍ Indiana (15)
❍ Iowa (16)
❍ Kansas (17)
❍ Kentucky (18)
❍ Louisiana (19)
❍ Maine (20)
❍ Maryland (21)
❍ Massachusetts (22)
❍ Michigan (23)
❍ Minnesota (24)
❍ Mississippi (25)
❍ Missouri (26)
❍ Montana (27)
❍ Nebraska (28)
❍ Nevada (29)
❍ New Hampshire (30)
❍ New Jersey (31)

| | | |
|---|---|---|
| 1146 | ○ | New Mexico (32) |
| 1147 | ○ | New York (33) |
| 1148 | ○ | North Carolina (34) |
| 1149 | ○ | North Dakota (35) |
| 1150 | ○ | Ohio (36) |
| 1151 | ○ | Oklahoma (37) |
| 1152 | ○ | Oregon (38) |
| 1153 | ○ | Pennsylvania (39) |
| 1154 | ○ | Rhode Island (40) |
| 1155 | ○ | South Carolina (41) |
| 1156 | ○ | South Dakota (42) |
| 1157 | ○ | Tennessee (43) |
| 1158 | ○ | Texas (44) |
| 1159 | ○ | Utah (45) |
| 1160 | ○ | Vermont (46) |
| 1161 | ○ | Virginia (47) |
| 1162 | ○ | Washington (48) |
| 1163 | ○ | West Virginia (49) |
| 1164 | ○ | Wisconsin (50) |
| 1165 | ○ | Wyoming (51) |
| 1166 | ○ | I do not live in the continental United States (52) |
| 1167 | | |
| 1168 | | Answer If Are you a resident of the United States? Yes Is Selected |
| 1169 | | Q3 What is your zip code? |
| 1170 | | |

| | |
|---|---|
| 1171 |  |
| 1172 | Q4 In which country do you reside? |
| 1173 | ○ Afghanistan (1) |
| 1174 | ○ Albania (2) |
| 1175 | ○ Algeria (3) |
| 1176 | ○ Andorra (4) |
| 1177 | ○ Angola (5) |
| 1178 | ○ Antigua and Barbuda (6) |
| 1179 | ○ Argentina (7) |
| 1180 | ○ Armenia (8) |
| 1181 | ○ Australia (9) |
| 1182 | ○ Austria (10) |
| 1183 | ○ Azerbaijan (11) |
| 1184 | ○ Bahamas (12) |
| 1185 | ○ Bahrain (13) |
| 1186 | ○ Bangladesh (14) |
| 1187 | ○ Barbados (15) |
| 1188 | ○ Belarus (16) |
| 1189 | ○ Belgium (17) |
| 1190 | ○ Belize (18) |
| 1191 | ○ Benin (19) |
| 1192 | ○ Bhutan (20) |
| 1193 | ○ Bolivia (21) |
| 1194 | ○ Bosnia and Herzegovina (22) |
| 1195 | ○ Botswana (23) |
| 1196 | ○ Brazil (24) |
| 1197 | ○ Brunei Darussalam (25) |
| 1198 | ○ Bulgaria (26) |
| 1199 | ○ Burkina Faso (27) |
| 1200 | ○ Burundi (28) |
| 1201 | ○ Cambodia (29) |
| 1202 | ○ Cameroon (30) |
| 1203 | ○ Canada (31) |
| 1204 | ○ Cape Verde (32) |
| 1205 | ○ Central African Republic (33) |
| 1206 | ○ Chad (34) |
| 1207 | ○ Chile (35) |
| 1208 | ○ China (36) |
| 1209 | ○ Colombia (37) |
| 1210 | ○ Comoros (38) |

❍ Congo, Republic of the... (39)
❍ Costa Rica (40)
❍ Côte d'Ivoire (41)
❍ Croatia (42)
❍ Cuba (43)
❍ Cyprus (44)
❍ Czech Republic (45)
❍ Democratic People's Republic of Korea (46)
❍ Democratic Republic of the Congo (47)
❍ Denmark (48)
❍ Djibouti (49)
❍ Dominica (50)
❍ Dominican Republic (51)
❍ Ecuador (52)
❍ Egypt (53)
❍ El Salvador (54)
❍ Equatorial Guinea (55)
❍ Eritrea (56)
❍ Estonia (57)
❍ Ethiopia (58)
❍ Fiji (59)
❍ Finland (60)
❍ France (61)
❍ Gabon (62)
❍ Gambia (63)
❍ Georgia (64)
❍ Germany (65)
❍ Ghana (66)
❍ Greece (67)
❍ Grenada (68)
❍ Guatemala (69)
❍ Guinea (70)
❍ Guinea-Bissau (71)
❍ Guyana (72)
❍ Haiti (73)
❍ Honduras (74)
❍ Hong Kong (S.A.R.) (75)
❍ Hungary (76)
❍ Iceland (77)
❍ India (78)

| 1251 | ○ Indonesia (79) |
| 1252 | ○ Iran, Islamic Republic of... (80) |
| 1253 | ○ Iraq (81) |
| 1254 | ○ Ireland (82) |
| 1255 | ○ Israel (83) |
| 1256 | ○ Italy (84) |
| 1257 | ○ Jamaica (85) |
| 1258 | ○ Japan (86) |
| 1259 | ○ Jordan (87) |
| 1260 | ○ Kazakhstan (88) |
| 1261 | ○ Kenya (89) |
| 1262 | ○ Kiribati (90) |
| 1263 | ○ Kuwait (91) |
| 1264 | ○ Kyrgyzstan (92) |
| 1265 | ○ Lao People's Democratic Republic (93) |
| 1266 | ○ Latvia (94) |
| 1267 | ○ Lebanon (95) |
| 1268 | ○ Lesotho (96) |
| 1269 | ○ Liberia (97) |
| 1270 | ○ Libyan Arab Jamahiriya (98) |
| 1271 | ○ Liechtenstein (99) |
| 1272 | ○ Lithuania (100) |
| 1273 | ○ Luxembourg (101) |
| 1274 | ○ Madagascar (102) |
| 1275 | ○ Malawi (103) |
| 1276 | ○ Malaysia (104) |
| 1277 | ○ Maldives (105) |
| 1278 | ○ Mali (106) |
| 1279 | ○ Malta (107) |
| 1280 | ○ Marshall Islands (108) |
| 1281 | ○ Mauritania (109) |
| 1282 | ○ Mauritius (110) |
| 1283 | ○ Mexico (111) |
| 1284 | ○ Micronesia, Federated States of... (112) |
| 1285 | ○ Monaco (113) |
| 1286 | ○ Mongolia (114) |
| 1287 | ○ Montenegro (115) |
| 1288 | ○ Morocco (116) |
| 1289 | ○ Mozambique (117) |
| 1290 | ○ Myanmar (118) |

| 1291 | ❍ Namibia (119) |
| 1292 | ❍ Nauru (120) |
| 1293 | ❍ Nepal (121) |
| 1294 | ❍ Netherlands (122) |
| 1295 | ❍ New Zealand (123) |
| 1296 | ❍ Nicaragua (124) |
| 1297 | ❍ Niger (125) |
| 1298 | ❍ Nigeria (126) |
| 1299 | ❍ North Korea (127) |
| 1300 | ❍ Norway (128) |
| 1301 | ❍ Oman (129) |
| 1302 | ❍ Pakistan (130) |
| 1303 | ❍ Palau (131) |
| 1304 | ❍ Panama (132) |
| 1305 | ❍ Papua New Guinea (133) |
| 1306 | ❍ Paraguay (134) |
| 1307 | ❍ Peru (135) |
| 1308 | ❍ Philippines (136) |
| 1309 | ❍ Poland (137) |
| 1310 | ❍ Portugal (138) |
| 1311 | ❍ Qatar (139) |
| 1312 | ❍ Republic of Korea (140) |
| 1313 | ❍ Republic of Moldova (141) |
| 1314 | ❍ Romania (142) |
| 1315 | ❍ Russian Federation (143) |
| 1316 | ❍ Rwanda (144) |
| 1317 | ❍ Saint Kitts and Nevis (145) |
| 1318 | ❍ Saint Lucia (146) |
| 1319 | ❍ Saint Vincent and the Grenadines (147) |
| 1320 | ❍ Samoa (148) |
| 1321 | ❍ San Marino (149) |
| 1322 | ❍ Sao Tome and Principe (150) |
| 1323 | ❍ Saudi Arabia (151) |
| 1324 | ❍ Senegal (152) |
| 1325 | ❍ Serbia (153) |
| 1326 | ❍ Seychelles (154) |
| 1327 | ❍ Sierra Leone (155) |
| 1328 | ❍ Singapore (156) |
| 1329 | ❍ Slovakia (157) |
| 1330 | ❍ Slovenia (158) |

| 1331 | ○ Solomon Islands (159) |
| --- | --- |
| 1332 | ○ Somalia (160) |
| 1333 | ○ South Africa (161) |
| 1334 | ○ South Korea (162) |
| 1335 | ○ Spain (163) |
| 1336 | ○ Sri Lanka (164) |
| 1337 | ○ Sudan (165) |
| 1338 | ○ Suriname (166) |
| 1339 | ○ Swaziland (167) |
| 1340 | ○ Sweden (168) |
| 1341 | ○ Switzerland (169) |
| 1342 | ○ Syrian Arab Republic (170) |
| 1343 | ○ Tajikistan (171) |
| 1344 | ○ Thailand (172) |
| 1345 | ○ The former Yugoslav Republic of Macedonia (173) |
| 1346 | ○ Timor-Leste (174) |
| 1347 | ○ Togo (175) |
| 1348 | ○ Tonga (176) |
| 1349 | ○ Trinidad and Tobago (177) |
| 1350 | ○ Tunisia (178) |
| 1351 | ○ Turkey (179) |
| 1352 | ○ Turkmenistan (180) |
| 1353 | ○ Tuvalu (181) |
| 1354 | ○ Uganda (182) |
| 1355 | ○ Ukraine (183) |
| 1356 | ○ United Arab Emirates (184) |
| 1357 | ○ United Kingdom of Great Britain and Northern Ireland (185) |
| 1358 | ○ United Republic of Tanzania (186) |
| 1359 | ○ United States of America (187) |
| 1360 | ○ Uruguay (188) |
| 1361 | ○ Uzbekistan (189) |
| 1362 | ○ Vanuatu (190) |
| 1363 | ○ Venezuela, Bolivarian Republic of... (191) |
| 1364 | ○ Viet Nam (192) |
| 1365 | ○ Yemen (193) |
| 1366 | ○ Zambia (580) |
| 1367 | ○ Zimbabwe (1357) |
| 1368 | |

| | |
|---|---|
| 1369 | Q5 Which best describes your gender |
| 1370 | ○ Male (1) |
| 1371 | ○ Female (2) |
| 1372 | ○ Prefer not to answer (3) |
| 1373 | |
| 1374 | Q6 What is your age? |
| 1375 | ○ 18-20 years (1) |
| 1376 | ○ 21-30 years (2) |
| 1377 | ○ 31-40 years (3) |
| 1378 | ○ 41-50 years (4) |
| 1379 | ○ 51-60 years (5) |
| 1380 | ○ 61-64 years (6) |
| 1381 | ○ 65 years and over (7) |
| 1382 | |
| 1383 | Q10 Which statement about beach visitation best describes your experience? |
| 1384 | ○ Infrequently (fewer than 10 times in my life) (1) |
| 1385 | ○ Once every year typically on vacation (2) |
| 1386 | ○ I go multiple times per year (3) |
| 1387 | ○ Several times per month (4) |
| 1388 | ○ Frequently (weekly or daily) (5) |
| 1389 | |
| 1390 | Q11 How would you describe the beaches that you commonly visit? |
| 1391 | ○ Calm with small to no waves (1) |
| 1392 | ○ Occasional wave activity, primarily during storms (2) |
| 1393 | ○ Regular wave activity but typically small or medium sized waves (3) |
| 1394 | ○ Strong waves are common (4) |
| 1395 | |
| 1396 | Q13 What is the main type of activity you do when you go to the beach? |
| 1397 | ○ Swimming and wading (1) |
| 1398 | ○ Board riding (including surfboard, boogie board, stand up, etc.) (2) |
| 1399 | ○ Beach activities only (sunbathing, shell collecting, etc.) (3) |
| 1400 | ○ Snorkeling or diving (4) |
| 1401 | ○ Other (5) |
| 1402 | |
| 1403 | Answer If What is the main type of activity you do when you go to the beach? Other Is Selected |
| 1404 | Q14 You answer other, please describe what you tend to do at the beach: |
| 1405 | |
| 1406 | Q16 Have you ever had swimming lessons or training, either in a pool or ocean? |
| 1407 | ○ Yes (1) |
| 1408 | ○ No (2) |
| 1409 | |

Q17 How would you rate your pool swimming ability?
○ unable to swim (1)
○ weak swimmer (2)
○ competent swimmer (3)
○ highly competent swimmer (4)
Q18 How far do you think you can swim in a pool before you have to stop/pause?
○ I can't swim (5)
○ Less than 25 yards (one length of a typical community swimming pool) (1)
○ More that 25 yards but less than 100 yards (2)
○ More than 100 yards but less than 500 yards (3)
○ More than 500 yards (4)
Q19 How would you rate your swimming ability in open water with waves (like an ocean or
lake)?
○ I have never swum in water with lots of waves (1)
○ Weak swimmer (2)
○ Competent swimmer (3)
○ Highly competent swimmer (4)
Q20 How far do you think you can swim in open water with waves before you have to
stop/pause?
○ Less than 25 yards (1)
○ More than 25 yards but less than 100 yards (2)
○ More than 100 yards but less than 500 yards (3)
○ More than 500 yards (4)
○ I can't swim (5)
Q21 Have you ever swum in an open ocean or lake with lots of wave breaking?
○ Yes (1)
○ No (2)
○ Unsure (3)

Q22 What is the most important factor for you when choosing an ocean or lake beach to visit,
with the intention of going into the water?
○ Safety (are not prone to theft, etc.) (1)
○ Safety in the water (avoid dangerous water hazards) (2)
○ Lifeguard presence (3)
○ Cleanliness of the beach and water (4)
○ Crowds (prefer to be on a popular beach) (5)
○ Crowds (prefer to be on a secluded, private or empty beach) (6)
○ Ease of access (7)
○ Avoid lots of breaking waves (i.e., prefer calm conditions) (8)
○ Prefer lots of breaking waves (9)
○ Other (10)
Answer If What is the most important factor for you when choosing an ocean or lake beach to visit,
with the intention of going into the water? Other Is Selected
Q23 You answered "other" to the previous questions.  Please describe the most important factor
for you when choosing an ocean or lake beach to visit:
Q24 When you go to the beach, how important is it to you to swim near a lifeguard?
○ Not important (1)
○ Important (2)
○ Very important (3)
Q25 If you visit a beach with no lifeguards, do you still go into the water to wade, swim or float?
○ Always (1)
○ Most of the Time (2)
○ Sometimes (3)
○ Rarely (4)
○ Never (5)
Q26 Do you think about or check for hazards when you go to the beach?
○ Always (1)
○ Most of the Time (2)
○ Sometimes (3)
○ Rarely (4)
○ Never (5)

Q27 What do you think is the most dangerous hazard when you swim, wade or float at the
beach?
○ Jellyfish (1)
○ Sharks (2)
○ Big waves (3)
○ Shorebreaks (4)
○ Undertow (5)
○ Alongshore currents (6)
○ Rip currents (7)
○ Surfboards/boogie boards/other swimmers (8)
○ Sunburn (9)
○ Other (10)
Answer If What do you think is the most dangerous hazard when you swim, wade or float at the
beach? Other Is Selected
Q28 You answered "other" to the previous question.  Please identify what you think is the most
dangerous hazard at the beach.
Q29 Have you ever seen or heard information about beach hazards.  Please select all that apply.
❑ Never (1)
❑ Yes, in primary school (2)
❑ Yes, in high school (3)
❑ Yes, at university/college (4)
❑ Yes, from my parents (5)
❑ Yes, through pamphlets and brochures (6)
❑ Yes, through warning signs on the beach (7)
❑ Yes, on the internet (8)
❑ Yes, on television (9)
❑ Yes, on the radio (10)
❑ Yes, at my rental property in the guide material (11)
❑ Other (12)
Answer If Have you ever seen or heard information about beach hazards.  Please select all
that apply. Other Is Selected
Q30 You answered "other' to the previous question.  Please describe where you have heard about
beach hazards.
Q31 Are you familiar with any beach safety flag system in the United States?
○ Yes (1)
○ No (2)

Q32 You answered "yes" to the previous question.  Can you describe what you know about the beach safety flag system in the United States?

Q35 Have you heard of rip currents?
- ○ Yes (1)
- ○ No (2)

Q37 Can you describe a rip current?

Q38 Where have you learned/heard about rip currents? Select all that apply.
- ❑ I have never heard of a rip current (1)
- ❑ Television (2)
- ❑ Magazine/book (3)
- ❑ Radio (4)
- ❑ Primary school (5)
- ❑ High school (6)
- ❑ College/University (7)
- ❑ Parents (8)
- ❑ Pamphlets and/or brochures (9)
- ❑ Internet (10)
- ❑ Beach signs (11)
- ❑ Lifeguard (12)
- ❑ I have been caught in one (direct experience) (13)
- ❑ Other (14)

Q39 You answered "other" to the previous question.  Please tell us where you have heard about rip currents.

Q40 If you were at a beach, would you know how to spot a rip current?
- ○ Yes, always (1)
- ○ Yes, sometimes (2)
- ○ No (3)
- ○ You can't see a rip current (4)
- ○ Unsure (5)

Q41 You answered "yes" to the previous question.  Can you describe what a rip current looks like?

Q42 Where on this photograph would you feel most safe to enter the water? Click on the picture at the spot along the beach that you believe is the safest.

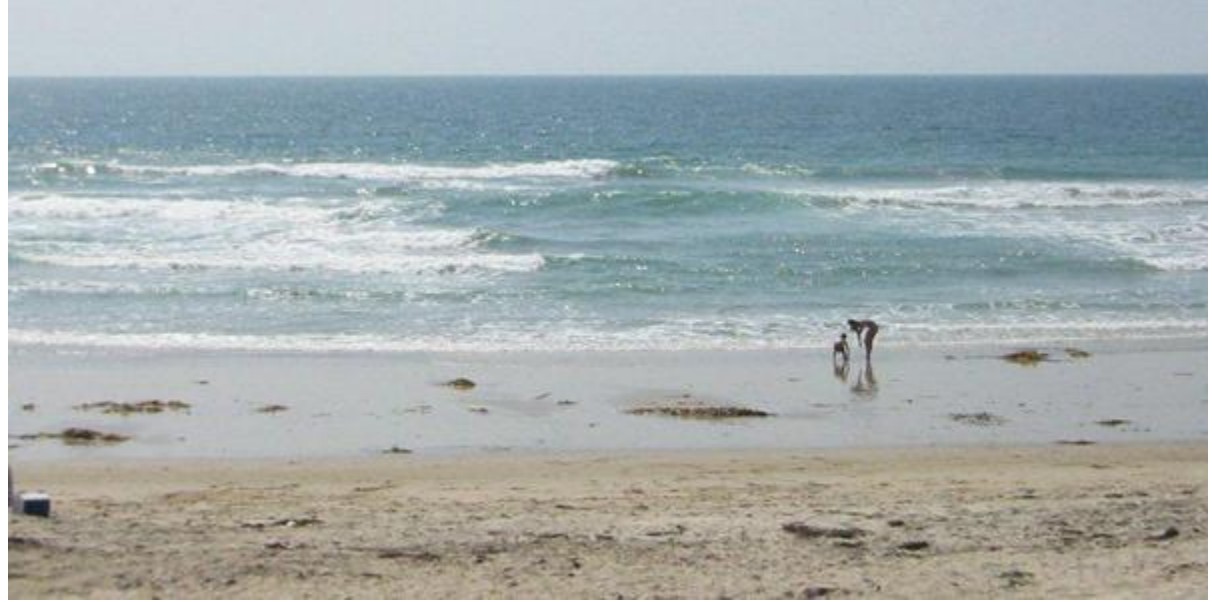

Q43 Where on this photograph would you feel most safe to enter the water? Click on the picture
at the spot along the beach that you believe is the safest.

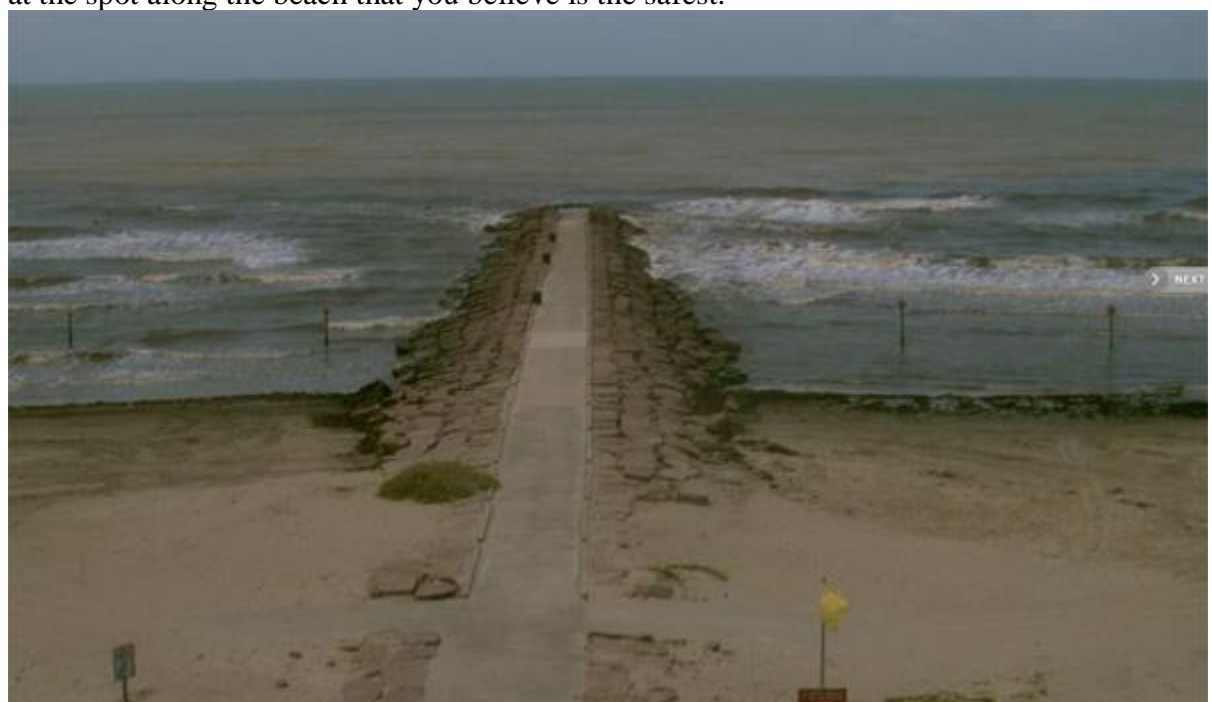


Q44 Where on this photograph would you feel most safe to enter the water? Click on the picture
at the spot along the beach that you believe is the safest.

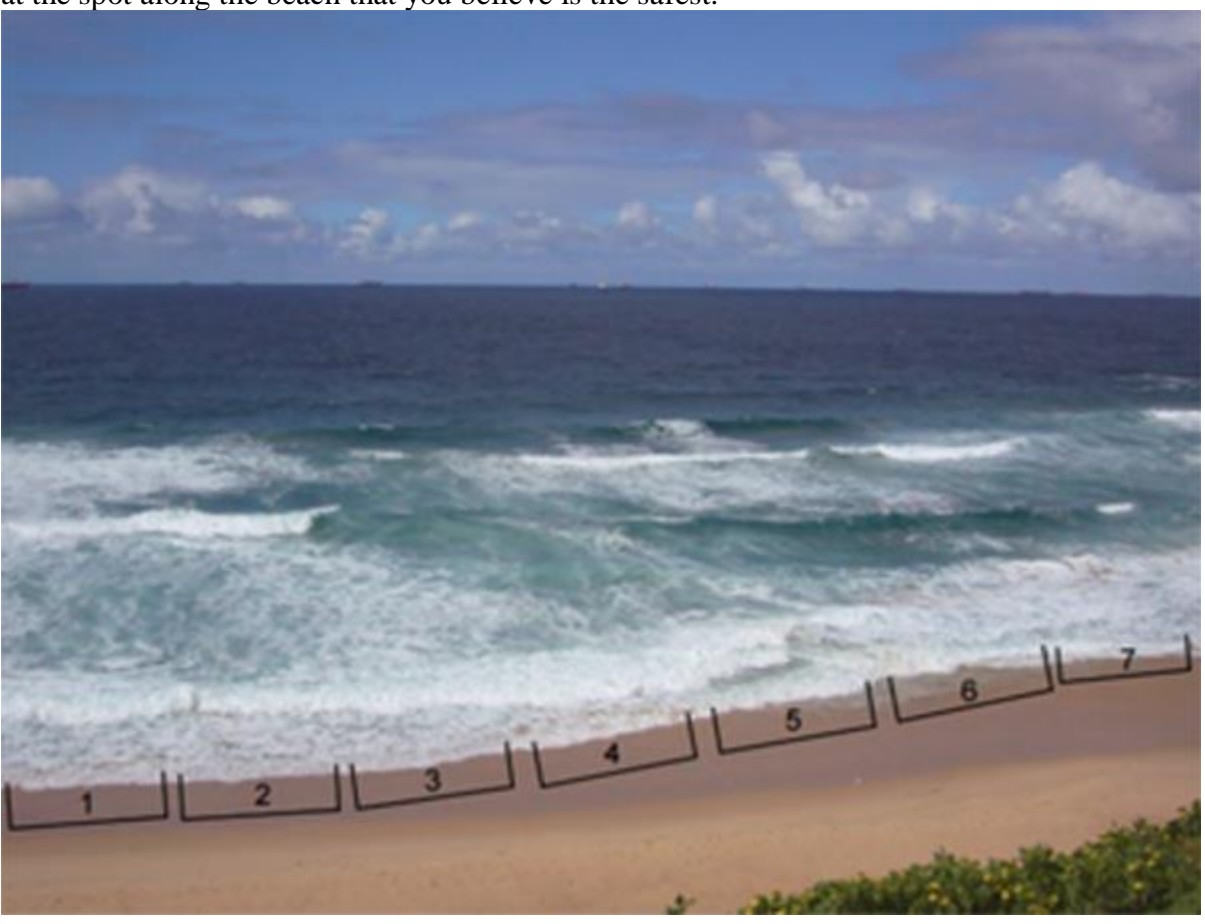

Q45 Explain what you should do if caught in a rip current?
Q46 Have you ever been caught in a rip current?
○   Yes, I was caught in a rip by accident (1)
○   Yes, I used the rip on purpose (e.g., for surfing) (2)
○   No (3)
○   Not sure (4)
Answer If Have you ever been caught in a rip current? Yes, I was caught in a rip by accident Is
Selected Or Have you ever been caught in a rip current? Yes, I used the rip on purpose (e.g. for
surfing) Is Selected
Q47 You answered that you had been caught in a rip current.  Where (ie. what beach) were you
caught in a rip current?

| 1592 |  |
| 1593 |  |
| 1594 | Q48 You answered that you were caught in a rip current by accident.  How did you get out of the |
| 1595 | rip current the first time this happened to you? |
| 1596 | ○ Self-escaped by swimming parallel to the beach first, then back to the beach (1) |
| 1597 | ○ Self-escaped by swimming straight back to the beach (2) |
| 1598 | ○ Self-escaped by just floating (3) |
| 1599 | ○ Rescued by lifeguard (4) |
| 1600 | ○ Rescued by bystander (e.g. family, friend, stranger, surfer) (5) |
| 1601 | ○ Don't know/can't remember (6) |
| 1602 | |
| 1603 | Q49 Before going to the beach, do you get information on the beach and surf conditions for the |
| 1604 | day? |
| 1605 | ○ Yes (1) |
| 1606 | ○ No (2) |
| 1607 | |
| 1608 |  |
| 1609 |  |
| 1610 | Q50 You answered "yes" to the previous question.  What source do you use to get information |
| 1611 | on the beach and surf conditions for the day? Select all that apply. |
| 1612 | ❑ Radio (1) |
| 1613 | ❑ Television (2) |
| 1614 | ❑ Internet (3) |
| 1615 | ❑ Facebook or other social media (4) |
| 1616 | ❑ Acquaintance (5) |
| 1617 | ❑ Other (6) |
| 1618 | |
| 1619 |  |
| 1620 |  |
| 1621 | Q51 You answered "other" to the previous question.  Please explain the other source of |
| 1622 | information about beach and surf conditions that you use. |
| 1623 | |
| 1624 |  |
| 1625 |  |
| 1626 | Q52 Does this information tend to affect your behavior when you go to the beach? |
| 1627 | ○ It doesn't affect my behavior (1) |
| 1628 | ○ It affects my behavior (2) |
| 1629 | |
| 1630 |  |
| 1631 |  |
| 1632 | Q53 Please explain how it affects your behavior at the beach. |
| 1633 | |

Q54 Rank the following sources of information from "most trusted" (1) to "least trusted" (5).
_____ Radio (1)
_____ Television (2)
_____ Internet (3)
_____ Facebook or other social media (4)
_____ Acquaintance (5)
Q55 Please explain why you trust one source of information more than another.
Q56 Have you ever seen beach safety information at the entrance to, or on beaches, that you
have visited?
❍ Yes (1)
❍ No (2)
Answer If Do you remember seeing any beach safety information at the entrance to the beach or on
the beach that you visit most often? Yes Is Selected
Q57 What type of beach safety information do you remember seeing?
❍ signs/posters (1)
❍ flags (2)
❍ pamphlets/brochures (3)
❍ other (4)
Answer If What type of beach safety information did you see? other Is Selected
Q58 You answered "other" to the previous question.  Please explain the type of beach safety
information that you tend to see at the entrance to the beach.
Answer If Do you remember seeing any beach safety information at the entrance to the beach or on
the beach that you visit most often? Yes Is Selected
Q59 Where do you tend to see the beach safety information?
❍ At the entrance to the beach (1)
❍ On the beach (2)
❍ Both on the beach and at the entrance to the beach (3)
Answer If Do you remember seeing any beach safety information at the entrance to the beach or on
the beach that you visit most often? Yes Is Selected
Q60 What is the primary message of the safety information that you tend to see?
Q61 Have you ever heard of the national United States rip current education campaign called
"Break the Grip of the Rip"©?
❍ Yes (1)
❍ No (2)

Answer If Have you ever heard of the "Break the Grip on the Rip" campaign? Yes Is Selected

Q62 You answered "yes" to the previous question.  Please tell us where you heard or have seen information related to the "Break the Grip of the Rip"© campaign.  Select all that apply.

- ❑ Radio (1)
- ❑ Television (2)
- ❑ Newspaper (3)
- ❑ Magazine/book (4)
- ❑ Local magazine or newspaper during my stay (5)
- ❑ Brochure/pamphlet (6)
- ❑ At my rental property here (7)
- ❑ Primary school (8)
- ❑ High school (9)
- ❑ College/University (10)
- ❑ Parents (11)
- ❑ Internet (12)
- ❑ "Break the Grip of the Rip"© website (13)
- ❑ Youtube or other internet video site (14)
- ❑ Facebook (15)
- ❑ Twitter (16)
- ❑ Other social media (17)
- ❑ Signs at the entrance to a beach (18)
- ❑ Signs on the beach (19)
- ❑ Lifeguards (20)
- ❑ Other (21)

Q63 What do you think "Break the Grip of the Rip"© means?

Sign Please use the following graphic when answering the next questions in the survey.

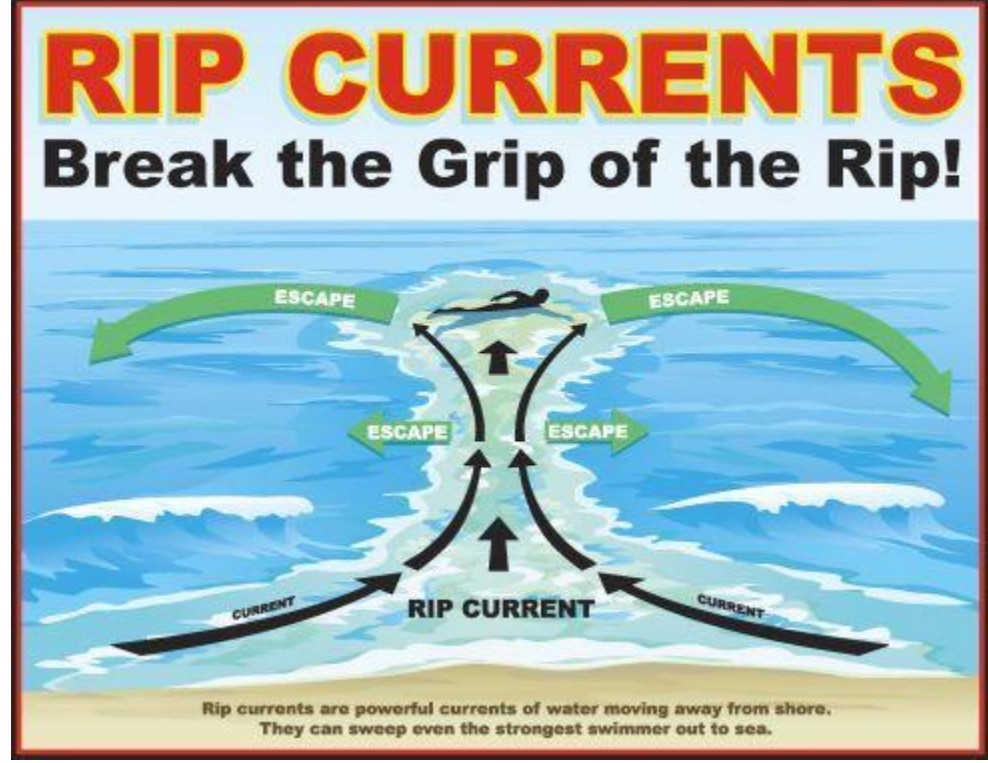

Q64 What does this sign tell you to do if caught in a rip current?
Q65 If you see this sign at a beach, how would it change your behavior at the beach?
Q66 Does this sign help you identify a rip current?
○ Yes (1)
○ No (2)
Answer If Does this sign help you identify a rip current? Yes Is Selected
Q67 You answered "yes" to the previous question. How does it help you identify a rip current?
Q68 What other information would be useful to be included in the "Break the Grip of the
Rip"© sign?

Q69 Have you ever seen or heard rip current forecasts from the following sources? Select all that
apply.
❑ Radio (1)
❑ Newspaper (2)
❑ Television (6)
❑ Internet (3)
❑ Social media (4)
❑ No (5)
Q70 Do you understand what it means if there is a "high risk" for rip currents?
○ Yes (1)
○ No (2)
Answer If Do you understand what it means if there is a "high risk" for rip currents? Yes Is Selected
Q71 You answered "yes" to the previous question.  What does a high risk of rip currents mean?
Q72 Do you understand what it means if there is a "low risk" for rip currents?
○ Yes (1)
○ No (2)
Q73 You answered "yes" to the previous question.  What does a low risk of rip currents mean?
Q74 Do you adjust your activities at the beach based on the rip forecast?
Q75 If you heard a rip current forecast (e.g. low risk or high risk) and went to the beach on the
same day, did the forecast match the conditions that you encountered at the beach?
○ Yes (1)
○ No (2)
Answer If you heard a rip current forecast (e.g. low risk or high risk) and went to the beach on the
same day, did the forecast match the conditions that you encountered at the beach? No Is Selected
Q76 You answered "no" to the previous question.  How did the conditions that you encountered
differ from the conditions that you experienced at the beach?