# Peer review of "Public Perceptions of a Rip Current Hazard Education Program: 'Break the Grip of the 2 Rip!' 3 4 5 Chris Houser1, Sarah Trimble2, Robert Brander3, B. Chris Brewster4, Greg Dusek5 Deborah Jones6, John Kuhn5 6 7 8 1</s"

_Natural Hazards and Earth System Sciences, 2017_

## Referee Comment (RC1) · Anonymous Referee #1 · 26 Feb 2017

The goal of the study was "to examine United States based beachgoers understanding of, and experience with the national Break the Grip of the Rip program and the rip current hazard in order to provide quantitative evidence to guide future improvements to beach safety education material".

Probably I'm the right person to comment this article because I have not experience with the kind of hazard analysed in the paper. I understand that the aim of the Authors is to analyse the results of the survey and that the article is not focused on the phenomenon "per se", nevertheless I think that they presume that all the readers know about it, while this is not true.

I.e., describing figure 2 they assume that all the readers know what are the most dangerous sectors, but it is not true instead (or it is not for me that only know Mediterranean

Sea and swimming pools). Maybe some notes in the caption of figure 2 could avoid that a reader having no experience with this type of phenomenon does not understand its importance and only can appreciate the correct scientific analysis of data.

The same impression reading the section Forecast. The Authors should first give clear information on the "right message", the right definition of high/low risk and then present the different people answers. In my opinion, this lack of information can generate confusion and obstruct a complete comprehension of the importance of the different answers.

Resuming, the paper can be improved taking in account two main problems that it presents:

1) The Authors, in my opinion, are too much focused on the results of their analysis and neglect to take into account that not all the readers know the analysed phenomenon.

2) The paper is very fluent but also very long and not schematic. I think that a further effort should be done to summarise the main results of each paragraph in a table for each paragraph, and also in a general table summarising all the findings in the discussion. Otherwise, as the paper is structured, the reader can not perceive each of the results obtained. Considering that this paper should be the starting point of an improvement of the Campaign, I think that the results should appear more clearly from the paper, in form of a list of bullets.

About figures:

Figure 2: The authors have the answer in mind but also the readers would like to know it.

Figure 3: some of the characters are impossible to read. I suggest reducing the description, reducing the size of the diagram, increasing the size of the characters and putting the labels vertically (print to understand if it is readable).

Figure 4: reduce the size of the diagram and increase the size of captions that currently

are impossible to read

Figure 5, 6 and 7: as for fig. 3

[Figure]

---

## Referee Comment (RC2) · R. Davidson-Arnott (Referee) · 10 Mar 2017

This paper presents results from an online survey of beach visitors in the USA which was directed at determining their perceptions of the 'break the grip of the rip" program specifically and, more generally, their knowledge of rip hazards and how to deal with them. The paper provides a useful introduction to the hazards posed by rip currents and the literature on this. It gives details of the break the grip program and also of related safety programs in place in the US to reduce drowning deaths related to rip currents. The methodology is clearly presented and illustrated with photographs and diagrams from the campaign and the questionnaire. The results are organised in sections around various themes which relate to the swimming ability and experience of the visitors with rip currents. These provide a useful means of evaluating the overall knowledge of rip currents and the hazards associated with them and also provide a
means of assessing future directions in terms of rip safety. There is, however, no section that focusses on familiarity with the "break the grip' program itself and it might be useful to tackle this first and then go on to the detailed analysis. The results section is a little lengthy and could be shortened a bit by confining the quotes to one or two per section since they are provided purely for illustration. The discussion is quite lengthy, but serves a useful purpose in drawing out the relevant messages from the survey itself and especially the contrast between frequent visitors, who were knowledgeable of the hazard, and infrequent visitors who were not knowledgeable and therefore likely to be most at risk. However, the key take-home messages in the discussion are not always apparent and it might be better to make them clearer in the conclusions by presenting them (the conclusions) as a set of concise bullet points that bring out the key results and recommendations rather than as a lengthy paragraph. The authors note in the introduction that the US has 4 coastlines (presumably the Arctic coast is omitted because of limited swimming opportunities) and that they differed considerably in terms of wave climate and beach systems. They also differ in the role of winds in generating or exacerbating the hazard. Thus, on the Great Lakes rip currents always occur in the presence of moderate to strong winds while on the west coast rip currents are often associated with large swell events and wind may be light. In the Great Lakes most rip current deaths appear to be associated with natural headlands, or with the presence of large groynes or harbour jetties but in Florida or Texas this is probably not the case. It might be useful therefore to comment on whether there were differences in responses based on which coast people used and to assess whether the education program should be tailored to individual coasts. Robin Davidson-Arnott University of Guelph

---

## Author Comment (AC1) · 4 Apr 2017

Reviewer #1

Comment 1: I understand that the aim of the Authors is to analyze the results of the survey and that the article is not focused on the phenomenon "per se", nevertheless I think that they presume that all the readers know about it, while this is not true. For example, describing figure 2 they assume that all the readers know what are the most dangerous sectors, but it is not true instead (or it is not for me that only know Mediterranean Sea and swimming pools). Maybe some notes in the caption of figure 2 could avoid that a reader having no experience with this type of phenomenon does not understand its importance and only can appreciate the correct scientific analysis of data.

[Figure]

*We agree with the reviewer that this level of detail is a needed revision to the manuscript. We will add notations to Figure 2 to identify the safe and unsafe areas in each picture, including the location of the rip current in each photograph. An additional annotation will be added to the Figure heading to let readers know that the annotation was not included in the original survey.

Comment 2:The same impression reading the section Forecast. The Authors should first give clear information on the "right message", the right definition of high/low risk and then present the different people answers. In my opinion, this lack of information can generate confusion and obstruct a complete comprehension of the importance of the different answers.

*The question raised by the reviewer represents one of the problems with the current warning systems for rips - there is no 'right message' for the definition of high or low risk. The forecast used by different agencies and in different areas are not consistent (as discussed on page 6, line 141), which means that it is not possible to identify the 'right message' for readers. However, we will add a statement to the methodology and results section on forecasts to remind the reader that there is no 'right message' and that we are only concerned about whether the respondent believed the message to be consistent with their observations.

Comment 3:The Authors, in my opinion, are too much focused on the results of their analysis and neglect to consider that not all the readers know the analyzed phenomenon.

*We will add a section in the introduction that describes rips in more detail and explain their formation. This will be combined with the suggestion by Reviewer #2 to describe how rip forcing and behavior may vary in different regions.

Comment 4:The paper is very fluent, but also very long and not schematic. I think that a further effort should be done to summarize the main results of each paragraph in a table for each paragraph, and also in a general table summarizing all the findings in the

discussion. Otherwise, as the paper is structured, the reader can not perceive each of the results obtained. Considering that this paper should be the starting point of an improvement of the Campaign, I think that the results should appear more clearly from the paper, in form of a list of bullets.

*This is a very interesting suggestion that will help to summarize the main findings from each section. We will add this table to the beginning of the discussion section. In response to Reviewer #2 we will also be modifying the conclusion section to include bulleted outcomes of the study.

Comment 5: Figure 2: The authors have the answer in mind but also the readers would like to know it.

*As noted above, we will add notations to Figure 2 to show the location of safe and unsafe swimming areas, as well as the location of the rip current in each photograph.

Comment 6: Figure 3: some of the characters are impossible to read. I suggest reducing the description, reducing the size of the diagram, increasing the size of the characters and putting the labels vertically (print to understand if it is readable).

*We will increase the size of the text in the revised manuscript to ensure that all characters are readable.

Comment 7: Figure 4: reduce the size of the diagram and increase the size of captions that currently are impossible to read

*We will increase the size of the text in the revised manuscript to ensure that all characters are readable.

Comment 8: Figure 5, 6 and 7: as for fig. 3

*We will increase the size of the text in the revised manuscript to ensure that all characters are readable.

Reviewer #2

This paper presents results from an online survey of beach visitors in the USA which was directed at determining their perceptions of the 'break the grip of the rip" program specifically and, more generally, their knowledge of rip hazards and how to deal with them. The paper provides a useful introduction to the hazards posed by rip currents and the literature on this. It gives details of the break the grip program and also of related safety programs in place in the US to reduce drowning deaths related to rip currents. The methodology is clearly presented and illustrated with photographs and diagrams from the campaign and the questionnaire. The results are organised in sections around various themes which relate to the swimming ability and experience of the visitors with rip currents. These provide a useful means of evaluating the overall knowledge of rip currents and the hazards associated with them and also provide a means of assessing future directions in terms of rip safety.

Comment 1: There is, however, no section that focusses on familiarity with the "break the grip' program itself and it might be useful to tackle this first and then go on to the detailed analysis.

*This is a valid criticism and we will add a section about the "break the grip' program at the start of the results section and use that as an introduction to the other results.

Comment 2: The results section is a little lengthy and could be shortened a bit by confining the quotes to one or two per section since they are provided purely for illustration.

*We included as many quotes as possible to ensure that we provided as much context and detail as possible for the readers. However, we recognize that there are large number of quotes and that they are only used for illustration. In this respect, we will reduce the number of quotes in the results section.

Comment 3: The discussion is quite lengthy, but serves a useful purpose in drawing out the relevant messages from the survey itself and especially the contrast between frequent visitors, who were knowledgeable of the hazard, and infrequent visitors who were not knowledgeable and therefore likely to be most at risk. However, the key takehome messages in the discussion are not always apparent and it might be better to make them clearer in the conclusions by presenting them (the conclusions) as a set of concise bullet points that bring out the key results and recommendations rather than as a lengthy paragraph.

*This is consistent with the comments of Reviewer #1, and we will include a table at the start of the discussion section to highlight the most important findings presented in the results section. We will also rewrite the conclusion section to be a set of bullets that summarizes the primary results of the study.

Comment 4: The authors note in the introduction that the US has 4 coastlines (presumably the Arctic coast is omitted because of limited swimming opportunities) and that they differed considerably in terms of wave climate and beach systems. They also differ in the role of winds in generating or exacerbating the hazard. Thus, on the Great Lakes rip currents always occur in the presence of moderate to strong winds while on the west coast rip currents are often associated with large swell events and wind may be light. In the Great Lakes, most rip current deaths appear to be associated with natural headlands, or with the presence of large groins or harbor jetties but in Florida or Texas this is probably not the case. It might be useful therefore to comment on whether there were differences in responses based on which coast people used and to assess whether the education program should be tailored to individual coasts.

*In response to Reviewer #1 we will be adding a paragraph to the introduction to describe rip currents and will use this section to describe the differences in the rip problem amongst the different coasts. While there is not enough information to determine whether location had an influence on the responses, we will add this as a qualifier and possible complicating factor in the discussion section.

Please also note the supplement to this comment:
http://www.nat-hazards-earth-syst-sci-discuss.net/nhess-2017-16/nhess-2017-16-AC1-supplement.pdf

---

## Author Response (AR1)

**Reviewer #1**

**I understand that the aim of the Authors is to analyse the results of the survey and that the article is not focused on the phenomenon "per se", nevertheless I think that they presume that all the readers know about it, while this is not true. For example, describing figure 2 they assume that all the readers know what are the most dangerous sectors, but it is not true instead (or it is not for me that only know Mediterranean Sea and swimming pools). Maybe some notes in the caption of figure 2 could avoid that a reader having no experience with this type of phenomenon does not understand its importance and only can appreciate the correct scientific analysis of data**.

[Figure]

- We agree with the reviewer that this level of detail is a needed in the revised manuscript. Specifically, we added notations to Figure 2 to identify the safe and unsafe areas in each picture, including the location of the rip current in each photograph.  An additional annotation will be added to the Figure heading to let readers know that the annotation was not included in the original survey.  As described further below, we have also added a section to the Introduction that describes rip currents in more detail. We have also added simple statements throughout the revised manuscript that help provide a basic understanding of rip currents.

*Fig. 2. Photographs used in Questions 42 through 44 of the survey to ask respondents "Where on this photograph would you swim?".  The location of the rip current in each photograph is shown by the red arrow, which was not visible to the respondents. In these examples, rip current location can also be identified by areas of reduced wave breaking.*

**The same impression reading the section Forecast. The Authors should first give clear information on the "right message", the right definition of high/low risk and then present the different people answers. In my opinion, this lack of information can generate confusion and obstruct a complete comprehension of the importance of the different answers.**

- The question raised by the reviewer represents one of the problems with the current warning systems for rips - there is no 'right message' for the definition of high or low risk. The forecast used by different agencies and in different areas are not consistent (as discussed on page 6, line 141 in the original manuscript), which means that it is not possible to identify the 'right message' for readers.  However, we have added several statements (see below) to the results section on forecasts to remind the reader that there is no 'right message' and that we are only concerned about whether the respondent believed the message to be consistent with their observations.

*…..The lack of consistency in forecasting is complicated by rip development being*
*dependent on how the incident wave field interacts with the pre-existing nearshore*
*morphology, which is difficult to predict without local knowledge on how it evolves over a*
*range of spatial and temporal scales.*
*Since perception of the rip hazard depends in part on trust in experts and authorities, and*
*trust in the protective measures they employ (Njome et al., 2010; Heitz et al., 2009;*
*Terpstra, 2009, 2011; Barnes, 2002), inaccuracies in the forecast or a discrepancy between*
*the forecast and what is observed at a specific beach at a specific time can erode*
*confidence in the forecast (Siegrist and Cvetkovich, 2000; Espluga et al., 2009). A lack of*
*confidence in the forecast could potentially condition beach users to downplay the hazard*
*warning on future visits (Hall and Slothower, 2009; Scolobig et al., 2012; Green et al.,*
*1991; Mileti and O'Brien, 1993)……*
*Respondents were also asked about whether they were aware of forecasts and whether*
*those forecasts altered their behavior, and if the forecasts conformed with their*
*observations at the beach. Since forecasts are not consistent and few are based on an*
*understanding of the pre-existing morphology, we were not worried about whether the*
*forecast was accurate, and focused on whether the respondent believed the message to*
*be consistent with their observations.*
*The problem lies in the fact that rip forecasts tend to be overly general to a larger region*
*and time and not necessarily dependent on an understanding of the pre-existing*
*morphology…*
*Moreover, it is difficult to predict the potential for rip development without an*
*understanding of the pre-existing nearshore morphology that is difficult to predict*
*without local knowledge on how it evolves over a range of spatial and temporal scales.*
**The Authors, in my opinion, are too much focused on the results of their analysis and neglect**
**to consider that not all the readers know the analysed phenomenon.**
• We have added a section in the introduction that describes rips in more detail and
explain their formation. This will be combined with the suggestion by Reviewer #2 to
describe how rip forcing and behavior may vary in different regions.
*Rip currents (often called "rips" or "rip tides") are strong, narrow seaward flows*
*driven by alongshore variations in wave breaking and resulting wave set-up landward of*
*the breaker zone. Due to their dependence on wave breaking, rips can develop in any*
*beach environment in oceanic, sea and lacustrine environments. Castelle et al. (2016)*
*classify rips as: 1) boundary rips that develop along both natural and engineered*
*structures including headlands, groins and piers, 2) bathymetric rips that develop in*
*response to variability of the nearshore morphology; and 3) hydrodynamic rips that are*

*spatially and temporally variable and develop in the absence of morphological variations*
*or a lateral boundary.  The type of rip that develops on a beach depends on the local wave*
*climate and geology. For example, rips in the Great Lakes tend to be associated with*
*natural headlands or the presence of large groins or harbor jetties, while rips in Florida*
*and Texas tend to be bathymetrically controlled and associated with a transverse bar and*
*rip nearshore morphology (Houser et al. 2013).   Rips also vary regionally based on the*
*driving forces, with rips on the Great Lakes typically associated with moderate to strong*
*winds, while on the West Coast of the United States, rips are often associated with large*
*swell events independent of the wind.*
*Rips are capable of carrying unsuspecting bathers significant distances away from*
*the shoreline with speeds reaching over 2 m s$^{-1}$.  As a consequence rips are considered a*
*major public health problem in the USA....*
**The paper is very fluent, but also very long and not schematic. I think that a further effort**
**should be done to summarise the main results of each paragraph in a table for each paragraph,**
**and also in a general table summarising all the findings in the discussion. Otherwise, as the**
**paper is structured, the reader can not perceive each of the results obtained. Considering that**
**this paper should be the starting point of an improvement of the Campaign, I think that the**
**results should appear more clearly from the paper, in form of a list of bullets.**
• This is a very good suggestion that will help to summarize the main findings from each
section.  We have added a table to the beginning of the discussion section.
**Table 2.** Summary of major findings from the "Break the Grip of the Rip!" National Rip Current
Survey.

| Focus of Questions | Example topics |
|---|---|
| Beach Preference | • Frequency and purpose of visits to a beach affect perception of surf conditions, importance of swimming near a lifeguard and self-reported ability to spot a rip current |
| Swimming Ability | • Range of self-reported swimming ability (distance in open water) related to self-reported competency |
| Ability to Identify a Rip Current | • Ability to identify safest location in a photograph related to frequency of beach visits, self-reported swimming competency and training
• Ability to identify safest location related to perceived importance of and concern about surf hazards, self-reported understanding of "high" and "low" risk conditions, and perceived accuracy of rip forecasts |

| | |
|---|---|
| Response to Warning Sign | • Perceived ability to use sign to identify a rip current varied with ability to identify safest location on a photograph
• Sign has been effective in communicating swimming parallel as an escape strategy, and taking caution when entering the water
• Identified need to provide a more accurate depiction of a rip current, detailed instructions on how to escape a rip current, and local emergency information |
| Prevention | • "Break the Grip of the Rip" Campaign has been successful in informing beach users to: 1) not fight the current; 2) swim out of the current, then to shore; 3) if you can't escape, float or tread water; and 4) if you need help, call or wave for assistance |
| Forecasts | • Self-reported change in behavior based on forecasted beach and surf conditions, but tendency for forecasts to be inconsistent with observations
• Perceived inaccuracy of forecast related to spatial and temporal broadness of forecast, inability to identify a rip, and behavior of other beach users |
| Trusted Sources of Information | • No significant correlations were observed between trust in a source of information and respondent demographics |

**Figure 2: The authors have the answer in mind but also the**
**readers would like to know it**.

• As noted above, we added notations to Figure 2 to
show the location of safe and unsafe swimming areas,
as well as the location of the rip current in each
photograph.

*Fig. 2. Photographs used in Questions 42 through 44*
*of the survey to ask respondents "Where on this*
*photograph would you swim?". The location of the rip*
*current in each photograph is shown by the red arrow,*
*which was not visible to the respondents.*

[Figure]

**Figure 3: some of the characters are impossible to read. I suggest reducing the description, reducing the size of the diagram, increasing the size of the characters and putting the labels vertically (print to understand if it is readable).**

- We have increased the size of the text in the revised manuscript to ensure that all characters are readable.

[Figure]

**Figure 4: reduce the size of the diagram and increase the size of captions that currently are impossible to read**

- We have increased the size of the text in the revised manuscript to ensure that all characters are readable.

[Figure]

 **Figure 5, 6 and 7: as for fig. 3**

· We have increased the size of the text in the revised manuscript to ensure that all
characters are readable.

[Figure]

[Figure]

[Figure]

**Reviewer #2**
**There is no section that focusses on familiarity with the "break the grip' program itself and it**
**might be useful to tackle this first and then go on to the detailed analysis.**
• This is a valid criticism and we have added a section about the "break the grip' program
at the start of the results section and use that as an introduction to the other results.
*3.1 Familiarity with the Break the Grip of the Rip ® Campaign*
*Only 18% (n=304) of respondents reported hearing about the Break the Grip of the Rip ®*
*Campaign with a nearly identical split by gender and age.  Of those who did,*
*approximately 40% reported hearing about the campaign either through a*
*brochure/pamphlet (n=120) or at the entrance to a beach (n=119). The majority of*
*respondents (54%; n=163) reported hearing about the campaign through various sources*
*on the internet including 90 respondents who reported having heard about the campaign*
*from the Break the Grip of the Rip ® website itself.  When asked what Break the Grip of*
*the Rip means, most respondents (familiar with the campaign) reported (to varying*
*degrees of accuracy) that it was designed to provide information about what to do if*
*caught in a rip current:*
          *Do not try to fight the current, instead work with the current*
                    *until you can break free of its pull*
    *Advises affected swimmers not to struggle while heading shoreward*
        *but to swim parallel to the beach till out of the off-beach current*
                *Swim parallel to get out of the rip*
*There were, however, several respondents (familiar with the campaign) who believed*
*that the messaging was not appropriate and needed to be rethought:*
          *The slogan is useless to anyone caught in a rip current!*
*What can you do by knowing this slogan? …."Wave, Yell & Swim Parallel"*
*is a far better slogan...it provides 3 lifesaving pieces of information. The existing slogan*
                        *provides nothing.*
      *it's an advertising slogan; it doesn't mean much at all.*
          *It's a bad slogan; it does not tell folks what to do,*
             *what to watch for, or anything useful.*
*Responses from those who were not familiar with the campaign were much shorter and*
*did not contain the level about survival strategies provided by those familiar with the*
*campaign.  Representative responses include "how to escape", "tips to survive", and "how*
*to get out of a rip".*

In support of this new section we added to the introduction and the discussion:
*Results from Brannstrom et al. (2015) suggest that while most beach users in Texas were*
*not familiar with the campaign itself, many were familiar with a key message of the*
*campaign on "what to do" when caught in a rip current. This suggests that the campaign*
*may have been successful in educating beach users and reducing the number of drownings,*
*but this hypothesis has never been formally tested.*
*Results of this rip current survey suggest that while many potential US beachgoers are not*
*aware of the "Break the Grip of the Rip" ® campaign, those that are tend to be informed*
*about rip current safety. While this is an encouraging result, it needs to be placed in*
*context.*
*It is also interesting to note that while many survey respondents were not familiar with the*
*"Break the Grip of the Rip" ® campaign itself, a clear majority of respondents (~91%)*
*understood the primary message of the campaign and were able to provide an explanation*
*of the message (i.e. "break the grip"), with those previously familiar with the campaign*
*providing detailed explanations of how to escape by 'swimming parallel' and/or 'floating*
*until the current weakened'. This also indicates that respondents may also have gained this*
*knowledge from other sources.*
**The results section is a little lengthy and could be shortened a bit by confining the quotes to**
**one or two per section since they are provided purely for illustration.**
• We included as many quotes as possible to ensure that we provided as much context and
detail as possible for the readers. However, we recognize that there are large number of
quotes and that they are only used for illustration. In this respect, we have reduced the
number of quotes in the results section by a third.
**The discussion is quite lengthy, but serves a useful purpose in drawing out the relevant**
**messages from the survey itself and especially the contrast between frequent visitors, who**
**were knowledgeable of the hazard, and infrequent visitors who were not knowledgeable and**
**therefore likely to be most at risk. However, the key take-home messages in the discussion are**
**not always apparent and it might be better to make them clearer in the conclusions by**
**presenting them (the conclusions) as a set of concise bullet points that bring out the key results**
**and recommendations rather than as a lengthy paragraph.**
• This is consistent with the comments of Reviewer #1, and we have therefore added a
table at the start of the discussion section to highlight the most important findings
presented in the results section. Because we have added this table, we maintained the
structure of the conclusion section with a broad summary and a focus on what can be
done to improve the campaign and forecasting.

**Table 2.** Summary of major findings from the "Break the Grip of the Rip!" National Rip Current
Survey.

| Focus of Questions | Example topics |
|---|---|
| Beach Preference | • Frequency and purpose of visits to a beach affect perception of surf conditions, importance of swimming near a lifeguard and self-reported ability to spot a rip current |
| Swimming Ability | • Range of self-reported swimming ability (distance in open water) related to self-reported competency |
| Ability to Identify a Rip Current | • Ability to identify safest location in a photograph related to frequency of beach visits, self-reported swimming competency and training
• Ability to identify safest location related to perceived importance of and concern about surf hazards, self-reported understanding of "high" and "low" risk conditions, and perceived accuracy of rip forecasts |
| Response to Warning Sign | • Perceived ability to use sign to identify a rip current varied with ability to identify safest location on a photograph
• Sign has been effective in communicating swimming parallel as an escape strategy, and taking caution when entering the water
• Identified need to provide a more accurate depiction of a rip current, detailed instructions on how to escape a rip current, and local emergency information |
| Prevention | • "Break the Grip of the Rip" Campaign has been successful in informing beach users to: 1) not fight the current, 2) swim out of the current, then to shore, 3) if you can't escape, float or tread water, and 4) if you need help, call or wave for assistance |
| Forecasts | • Self-reported change in behavior based on forecasted beach and surf conditions, but tendency for forecasts to be inconsistent with observations
• Perceived inaccuracy of forecast related to spatial and temporal broadness of forecast, inability to identify a rip, and behavior of other beach users |
| Trusted Sources of Information | • No significant correlations were observed between trust in a source of information and respondent demographics |

**The authors note in the introduction that the US has 4 coastlines (presumably the Arctic coast is omitted because of limited swimming opportunities) and that they differed considerably in terms of wave climate and beach systems. They also differ in the role of winds in generating or exacerbating the hazard. Thus, on the Great Lakes rip currents always occur in the presence of moderate to strong winds while on the west coast rip currents are often associated with large swell events and wind may be light. In the Great Lakes most rip current deaths appear to be associated with natural headlands, or with the presence of large groynes or harbour jetties but in Florida or Texas this is probably not the case. It might be useful therefore to comment on whether there were differences in responses based on which coast people used and to assess whether the education program should be tailored to individual coasts.**

- In response to Reviewer #1 we will add to the introduction to describe rip currents and will use this section to describe the differences in the rip problem amongst the different coasts. While there is not enough information to determine whether location had an influence on the responses, we will add this as a qualifier and possible complicating factor in the discussion section.

  *Rip currents (often called "rips" or "rip tides") are strong, narrow seaward flows driven by alongshore variations in wave set-up landward of the breaker zone. Due to their dependence on wave breaking, rips can develop in any beach environment in oceanic, sea and lacustrine environments. Castelle et al. (2016) classify rips as: 1) boundary rips that develop along both natural and engineered structures including headlands, groins and piers, 2) bathymetric rips that develop in response to the variability of the nearshore morphology and 3) hydrodynamic rips that are spatial and temporally variable and develop in the absence of morphological variations or a lateral boundary. The type of rip that develops on a beach depends on the local wave climate and geology. For example, rips in the Great Lakes tend to be associated with natural headlands or the presence of large groins or harbor jetties, while rips in Florida and Texas tend to be bathymetrically controlled and associated with a transverse bar and rip nearshore morphology (Houser et al. 2013). Rips also vary regionally based on the driving forces, with rips on the Great Lakes typically associated with moderate to strong winds, while on the West Coast of the United States the rips are often associated with large swell events independent of the wind.*

  *Rips are capable of carrying unsuspecting bathers significant distances away from the shoreline with speeds reaching over 2 m s$^{-1}$. As a consequence rips are considered a major public health problem in the USA….*

*Finally, we have made small edits throughout the manuscript in an attempt to reduce the overall length of the paper without compromising the content and findings.*

[revised manuscript text omitted]

Answer If How does the information from this site affect your behavior at the beach? If affects my behavior Is Selected

Q53 Please explain how it affects your behavior at the beach.

Q54 Rank the following sources of information from "most trusted" (1) to "least trusted" (5).
____ Radio (1)
____ Television (2)
____ Internet (3)
____ Facebook or other social media (4)
____ Acquaintance (5)
Q55 Please explain why you trust one source of information more than another.
Q56 Have you ever seen beach safety information at the entrance to, or on beaches, that you
have visited?
○ Yes (1)
○ No (2)
Answer If Do you remember seeing any beach safety information at the entrance to the beach or on
the beach that you visit most often? Yes Is Selected
Q57 What type of beach safety information do you remember seeing?
○ signs/posters (1)
○ flags (2)
○ pamphlets/brochures (3)
○ other (4)
Answer If What type of beach safety information did you see? other Is Selected
Q58 You answered "other" to the previous question.  Please explain the type of beach safety
information that you tend to see at the entrance to the beach.
Answer If Do you remember seeing any beach safety information at the entrance to the beach or on
the beach that you visit most often? Yes Is Selected
Q59 Where do you tend to see the beach safety information?
○ At the entrance to the beach (1)
○ On the beach (2)
○ Both on the beach and at the entrance to the beach (3)
Answer If Do you remember seeing any beach safety information at the entrance to the beach or on
the beach that you visit most often? Yes Is Selected
Q60 What is the primary message of the safety information that you tend to see?
Q61 Have you ever heard of the national United States rip current education campaign called
"Break the Grip of the Rip"©?
○ Yes (1)
○ No (2)

Answer If Have you ever heard of the "Break the Grip on the Rip" campaign? Yes Is Selected

Q62 You answered "yes" to the previous question.  Please tell us where you heard or have seen information related to the "Break the Grip of the Rip"© campaign.  Select all that apply.

❑ Radio (1)

❑ Television (2)

❑ Newspaper (3)

❑ Magazine/book (4)

❑ Local magazine or newspaper during my stay (5)

❑ Brochure/pamphlet (6)

❑ At my rental property here (7)

❑ Primary school (8)

❑ High school (9)

❑ College/University (10)

❑ Parents (11)

❑ Internet (12)

❑ "Break the Grip of the Rip"© website (13)

❑ Youtube or other internet video site (14)

❑ Facebook (15)

❑ Twitter (16)

❑ Other social media (17)

❑ Signs at the entrance to a beach (18)

❑ Signs on the beach (19)

❑ Lifeguards (20)

❑ Other (21)

Q63 What do you think "Break the Grip of the Rip"© means?

Sign Please use the following graphic when answering the next questions in the survey.

[Figure]

Q64 What does this sign tell you to do if caught in a rip current?

Q65 If you see this sign at a beach, how would it change your behavior at the beach?

Q66 Does this sign help you identify a rip current?
○ Yes (1)
○ No (2)

Answer If Does this sign help you identify a rip current? Yes Is Selected
Q67 You answered "yes" to the previous question.  How does it help you identify a rip current?

Q68 What other information would be useful to be included in the "Break the Grip of the
Rip"© sign?

Q69 Have you ever seen or heard rip current forecasts from the following sources? Select all that
apply.
❑ Radio (1)
❑ Newspaper (2)
❑ Television (6)
❑ Internet (3)
❑ Social media (4)
❑ No (5)
Q70 Do you understand what it means if there is a "high risk" for rip currents?
○ Yes (1)
○ No (2)
Answer If Do you understand what it means if there is a "high risk" for rip currents? Yes Is Selected
Q71 You answered "yes" to the previous question.  What does a high risk of rip currents mean?
Q72 Do you understand what it means if there is a "low risk" for rip currents?
○ Yes (1)
○ No (2)
Q73 You answered "yes" to the previous question.  What does a low risk of rip currents mean?
Q74 Do you adjust your activities at the beach based on the rip forecast?
Q75 If you heard a rip current forecast (e.g. low risk or high risk) and went to the beach on the
same day, did the forecast match the conditions that you encountered at the beach?
○ Yes (1)
○ No (2)
Answer If you heard a rip current forecast (e.g. low risk or high risk) and went to the beach on the
same day, did the forecast match the conditions that you encountered at the beach? No Is Selected
Q76 You answered "no" to the previous question.  How did the conditions that you encountered
differ from the conditions that you experienced at the beach?

---

## Referee Report (RR1)

The Authors improved the paper by adding a description of the phenomenon analysed and a clear description of rip evidences in pictures. Nevertheless, some formatting features could be applied to strengthen the message and be more concise.

1. the diagrams seems to have the same problems of the previous version. On my monitor, I cannot read the labels of the column in histograms. It is sufficient to reduce the size of the diagram and increase the size of the labels.

2. it is important to add the appendix to the paper because it explain the survey steps and can contain suggestions for similar surveys, even focused on different topics. Nevertheless, the Authors should do a minimum effort to rearrange it in a concise, pleasant and printable way, reducing it into a table, in order to fill the entire page by putting sessions in two columns, for example.

3. I also suggest checking formatting criteria especially of the parts in the text where they quote the answers of respondents. The format applied, could be changed in a more concise way, and being sure that the spaces before and after the quotations are always the same throughout the entire text.

An example:

| 497 | inconsistency reflected the temporal and spatial broadness of the rip forecast compared to what |
| 498 | they observed: |
| 499 | |
| 500 | |
| 501 | *Weather changed quickly and no beach flags were posted, advising of rip* |
| 502 | *currents.* |
| 503 | |
| 504 | |
| 505 | |
| 506 | *Rip currents cannot be predicted for individual beaches, they are blanket* |
| 507 | *warnings.* |
| 508 | |
| 509 | Other respondents noted the forecast was inaccurate because other beach users had not adjusted |
| 510 | their behavior: |
| 511 | |
| 512 | *I never noticed an[y] thing unusual and people in general don't seem to adjust* |
| 513 | *their behavior.* |
| 514 | |
| 515 | |
| 516 | Others noted it was not possible to determine if the forecast was accurate because they were not |
| 517 | able to spot a rip on the beach at that specific time or in general: |

inconsistency reflected the temporal and spatial broadness of the rip forecast compared to what 497 they observed:

➢ *Weather changed quickly and no beach flags were posted, advising of rip currents*
➢ *Rip currents cannot be predicted for individual beaches, they are blanket warnings.*

Other respondents noted the forecast was inaccurate because other beach users had not adjusted their behavior:

➢ *I never noticed an[y] thing unusual and people in general don't seem to adjust their behavior.*

Others noted it was not possible to determine if the forecast was accurate because they were not 516 able to spot a rip on the beach at that specific time or in general:

---

## Author Response (AR2)

**Managing Editor**

The diagrams seem to have the same problems of the previous version. On my monitor, I cannot read the labels of the column in histograms. It is sufficient to reduce the size of the diagram and increase the size of the labels.

- This was a conversion issue with the PDF maker online. We have removed all figures from the document and saved independently as TIFF and uploaded as a zip file. There should be no issues with the figures individually saved and not as part of the PDF.

It is important to add the appendix to the paper because it explain the survey steps and can contain suggestions for similar surveys, even focused on different topics. Nevertheless, the Authors should do a minimum effort to rearrange it in a concise, pleasant and printable way, reducing it into a table, in order to fill the entire page by putting sessions in two columns, for example.

- We have transferred the survey instrument to a 2-column table

**Appendix**

Survey Instrument

| Question | Response |
| --- | --- |
| Are you a resident of the United States? | Yes/No |
| If resident- in what state do you currently reside? | List of US States |
| If resident- What is your zip code? | Open response |
| If not resident- In which country do you reside? | List of Countries |
| Which best describes your gender? | Male/Female/No response |
| What is your Age? | ☐ 18-20 years
☐ 21-30 years
☐ 31-40 years
☐ 41-50 years
☐ 51-60 years
☐ 61-64 years
☐ 65 years and over |
| Which statement about beach visitation best describes your experience? | ☐ Infrequently (fewer than 10 times in my life)
☐ Once every year typically on vacation
☐ I go multiple times per year
☐ Several times per month
☐ Frequently (weekly or daily) |
| How would you describe the beaches that you commonly visit? | ☐ Calm with small to no waves
☐ Occasional wave activity, primarily during storms
☐ Regular wave activity but typically small or medium sized waves
☐ Strong waves are common |
| What is the main type of activity you do when you go to the beach? | ☐ Swimming and wading
☐ Board riding (including surfboard, boogie board, stand up, etc.)
☐ Beach activities only (sunbathing, shell collecting, etc.)
☐ Snorkeling or diving
☐ Other |
| If Other selected- describe what you tend to do at the beach. | Open Response |
| Have you ever had swimming lessons or | Yes/No |

I also suggest checking formatting criteria especially of the parts in the text where they quote the answers of respondents. The format applied, could be changed in a more concise way, and being sure that the spaces before and after the quotations are always the same throughout the entire text.

- As suggested we made all quotes as bullets and made sure that all spaces are the same throughout the document using the example provided

There were, however, several respondents (familiar with the campaign) who believed that the messaging was not appropriate and needed to be rethought:

> *The slogan is useless to anyone caught in a rip current! What can you do by knowing this slogan? ...."Wave, Yell & Swim Parallel" is a far better slogan...it provides 3 lifesaving pieces of information. The existing slogan provides nothing.*

> *It's an advertising slogan; it doesn't mean much at all. It's a bad slogan; it does not tell folks what to do, what to watch for, or anything useful.*

Responses from those who were not familiar with the campaign were much shorter and did not contain the level about survival strategies provided by those familiar with the campaign. Representative responses include "how to escape", "tips to survive", and "how to get out of a rip".